# Ridge Regression: Structure, Cross-Validation, and Sketching

**Sifan Liu**
Department of Statistics
Stanford University
Stanford, CA 94305, USA
sfliu@stanford.edu

**Edgar Dobriban**
Department of Statistics
University of Pennsylvania
Philadelphia, PA 19104, USA
dobriban@wharton.upenn.edu

## Abstract

We study the following three fundamental problems about ridge regression: (1) what is the structure of the estimator? (2) how to correctly use cross-validation to choose the regularization parameter? and (3) how to accelerate computation without losing too much accuracy? We consider the three problems in a unified large-data linear model. We give a precise representation of ridge regression as a covariance matrix-dependent linear combination of the true parameter and the noise. We study the bias of $K$-fold cross-validation for choosing the regularization parameter, and propose a simple bias-correction. We analyze the accuracy of primal and dual sketching for ridge regression, showing they are surprisingly accurate. Our results are illustrated by simulations and by analyzing empirical data.

## 1 Introduction

Ridge or $\ell_2$-regularized regression is a widely used method for prediction and estimation when the data dimension $p$ is large compared to the number of datapoints $n$. This is especially so in problems with many good features, where sparsity assumptions may not be justified. A great deal is known about ridge regression. It is Bayes optimal for any quadratic loss in a Bayesian linear model where the parameters and noise are Gaussian. The asymptotic properties of ridge have been widely studied (e.g., Tulino & Verdú, 2004; Serdobolskii, 2007; Couillet & Debbah, 2011; Dicker, 2016; Dobriban & Wager, 2018, etc). For choosing the regularization parameter in practice, cross-validation (CV) is widely used. In addition, there is an exact shortcut (e.g., Hastie et al., 2009, p. 243), which has good consistency properties (Hastie et al., 2019). There is also a lot of work on fast approximate algorithms for ridge, e.g., using sketching methods (e.g., el Alaoui & Mahoney, 2015; Chen et al., 2015; Wang et al., 2018; Chowdhury et al., 2018, among others).

Here we seek to develop a deeper understanding of ridge regression, going beyond existing work in multiple aspects. We work in linear models under a popular asymptotic regime where $n, p \to \infty$ at the same rate (Marchenko & Pastur, 1967; Serdobolskii, 2007; Couillet & Debbah, 2011; Yao et al., 2015). In this framework, we develop a fundamental representation for ridge regression, which shows that it is well approximated by a linear scaling of the true parameters perturbed by noise. The scaling matrices are functions of the population-level covariance of the features. As a consequence, we derive formulas for the training error and bias-variance tradeoff of ridge.

Second, we study commonly used methods for choosing the regularization parameter. Inspired by the observation that CV has a bias for estimating the error rate (e.g., Hastie et al., 2009, p. 243), we study the bias of CV for selecting the regularization parameter. We discover a surprisingly simple form for the bias, and propose a downward scaling bias correction procedure. Third, we study the accuracy loss of a class of randomized sketching algorithms for ridge regression. These algorithms approximate the sample covariance matrix by sketching or random projection. We show they can be surprisingly accurate, e.g., they can sometimes cut computational cost in half, only incurring 5% extra error. Even more, they can sometimes improve the MSE if a suboptimal regularization parameter is originally used.

Our work leverages recent results from asymptotic random matrix theory and free probability theory. One challenge in our analysis is to find the limit of the trace $\operatorname{tr}(\Sigma_1 + \Sigma_2^{-1})^{-1}/p$, where $\Sigma_1$ and $\Sigma_2$ are $p \times p$ independent sample covariance matrices of Gaussian random vectors. The calculation requires nontrivial aspects of freely additive convolutions (e.g., Voiculescu et al., 1992; Nica & Speicher, 2006).

Our work is connected to prior works on ridge regression in high-dimensional statistics (Serdobolskii, 2007) and wireless communications (Tulino & Verdú, 2004; Couillet & Debbah, 2011). Among other related works, El Karoui & Kösters (2011) discuss the implications of the geometric sensitivity of random matrix theory for ridge regression, without considering our problems. El Karoui (2018) and Dicker (2016) study ridge regression estimators, but focus only on the risk for identity covariance. Hastie et al. (2019) study "ridgeless" regression, where the regularization parameter tends to zero.

Sketching is an increasingly popular research topic, see Vempala (2005); Halko et al. (2011); Mahoney (2011); Woodruff (2014); Drineas & Mahoney (2017) and references therein. For sketched ridge regression, Zhang et al. (2013a;b) study the dual problem in a complementary finite-sample setting, and their results are hard to compare. Chen et al. (2015) propose an algorithm combining sparse embedding and the subsampled randomized Hadamard transform (SRHT), proving relative approximation bounds. Wang et al. (2017) study iterative sketching algorithms from an optimization point of view, for both the primal and the dual problems. Dobriban & Liu (2018) study sketching using asymptotic random matrix theory, but only for unregularized linear regression. Chowdhury et al. (2018) propose a data-dependent algorithm in light of the ridge leverage scores. Other related works include Sarlos (2006); Ailon & Chazelle (2006); Drineas et al. (2006; 2011); Dhillon et al. (2013); Ma et al. (2015); Raskutti & Mahoney (2016); Gonen et al. (2016); Thanei et al. (2017); Ahfock et al. (2017); Lopes et al. (2018); Huang (2018).

The structure of the paper is as follows: We state our results on representation, risk, and bias-variance tradeoff in Section 2. We study the bias of cross-validation for choosing the regularization parameter in Section 3. We study the accuracy of randomized primal and dual sketching for both orthogonal and Gaussian sketches in Section 4. We provide proofs and additional simulations in the Appendix. Code reproducing the experiments in the paper are available at `https://github.com/liusf15/RidgeRegression`.

## 2 RIDGE REGRESSION

We work in the usual linear regression model $Y = X\beta + \varepsilon$, where each row $x_i$ of $X \in \mathbb{R}^{n \times p}$ is a datapoint in $p$ dimensions, and so there are $p$ features. The corresponding element $y_i$ of $Y \in \mathbb{R}^n$ is its continous response (or outcome). We assume mean zero uncorrelated noise, so $\mathbb{E}\varepsilon = 0$, and $\operatorname{Cov}[\varepsilon] = \sigma^2 I_n$. We estimate the coefficient $\beta \in \mathbb{R}^p$ by ridge regression, solving the optimization problem

$$\hat{\beta} = \arg \min_{\beta \in \mathbb{R}^p} \frac{1}{n}\|Y - X\beta\|_2^2 + \lambda\|\beta\|_2^2,$$

where $\lambda > 0$ is a regularization parameter. The solution has the closed form

$$\hat{\beta} = \left(X^\top X/n + \lambda I_p\right)^{-1} X^\top Y/n. \tag{1}$$

We work in a "big data" asymptotic limit, where both the dimension $p$ and the sample size $n$ tend to infinity, and their aspect ratio converges to a constant, $p/n \to \gamma \in (0, \infty)$. Our results can be interpreted for any $n$ and $p$, using $\gamma = p/n$ as an approximation.

We recall that the empirical spectral distribution (ESD) of a $p \times p$ symmetric matrix $\Sigma$ is the distribution $\frac{1}{p}\sum_{i=1}^{p} \delta_{\lambda_i}$ where $\lambda_i$, $i = 1, \ldots, p$ are the eigenvalues of $\Sigma$, and $\delta_x$ is the point mass at $x$. This is a standard notion in random matrix theory, see e.g., Marchenko & Pastur (1967); Tulino & Verdú (2004); Couillet & Debbah (2011); Yao et al. (2015). The ESD is a convenient tool to summarize all information obtainable from the eigenvalues of a matrix. For instance, the trace of $\Sigma$ is proportional to the *mean* of the distribution, while the condition number is related to the *range of the support*. As is common, we will work in models where there is a sequence of covariance matrices $\Sigma = \Sigma_p$, and their ESDs converges in distribution to a limiting probability distribution. The results become simpler, because they depend only on the limit.

By extension, we say that the ESD of the $n \times p$ matrix $X$ is the ESD of $X^\top X/n$. We will consider some very specific models for the data, assuming it is of the form $X = U\Sigma^{1/2}$, where $U$ has iid entries of zero mean and unit variance. This means that the datapoints, i.e., the rows of $X$, have the form $x_i = \Sigma^{1/2} u_i$, $i = 1, \ldots, p$, where $u_i$ have iid entries. Then $\Sigma$ is the "true" covariance matrix of the features, which is typically not observed. These types of models for the data are very common in random matrix theory, see the references mentioned above.

Under these models, it is possible to characterize precisely the deviations between the empirical covariance matrix $\widehat{\Sigma} = n^{-1}X^\top X$ and the population covariance matrix $\Sigma$, dating back to the well known classical Marchenko-Pastur law for eigenvectors (Marchenko & Pastur, 1967), extended to more general models and made more precise, including results for eigenvectors (see e.g., Tulino & Verdú, 2004; Couillet & Debbah, 2011; Yao et al., 2015, and references therein). This has been used to study methods for estimating the true covariance matrix, with several applications (e.g., Paul & Aue, 2014; Bun et al., 2017). More recently, such models have been used to study high dimensional statistical learning problems, including classification and regression (e.g., Zollanvari & Genton, 2013; Dobriban & Wager, 2018). Our work falls in this line.

We start by finding a precise representation of the ridge estimator. For random vectors $u_n, v_n$ of growing dimension, we say $u_n$ and $v_n$ are *deterministic equivalents*, if for any sequence of fixed (or random and independent of $u_n, v_n$) vectors $w_n$ such that $\limsup \|w_n\|_2 < \infty$ almost surely, we have $|w_n^\top(u_n - v_n)| \to 0$ almost surely. We denote this by $u_n \asymp v_n$. Thus linear combinations of $u_n$ are well approximated by those of $v_n$. This is a somewhat non-standard definition, but it turns out that it is precisely the one we need to use prior results from random matrix theory such as from (Rubio & Mestre, 2011).

We extend scalar functions $f : \mathbb{R} \to \mathbb{R}$ to matrices in the usual way by functional calculus, applying them to the eigenvalues and keeping the eigenvectors. If $M = V\Lambda V^\top$ is a spectral decomposition of $M$, then we define $f(M) := V f(\Lambda) V^\top$, where $f(\Lambda)$ is the diagonal matrix with entries $f(\Lambda_{ii})$.

For a fixed design matrix $X$, we can write the estimator as

$$\hat{\beta} = (\widehat{\Sigma} + \lambda I_p)^{-1}\widehat{\Sigma}\beta + (\widehat{\Sigma} + \lambda I_p)^{-1}\frac{X^\top \varepsilon}{n}.$$

However, for a random design, we can find a representation that depends on the true covariance $\Sigma$, which may be simpler when $\Sigma$ is simple, e.g., when $\Sigma = I_p$ is isotropic.

**Theorem 2.1** (Representation of ridge estimator). *Suppose the data matrix has the form $X = U\Sigma^{1/2}$, where $U \in \mathbb{R}^{n \times p}$ has iid entries of zero mean, unit variance and finite $8 + c$-th moment for some $c > 0$, and $\Sigma = \Sigma_p \in \mathbb{R}^{p \times p}$ is a deterministic positive definite matrix. Suppose that $n, p \to \infty$ with $p/n \to \gamma > 0$. Suppose the ESD of the sequence of $\Sigma$s converges in distribution to a probability measure with compact support bounded away from the origin. Suppose that the noise is Gaussian, and that $\beta = \beta_p$ is an arbitrary sequence of deterministic vectors, such that $\limsup \|\beta\|_2 < \infty$.*

*Then the ridge regression estimator is asymptotically equivalent to a random vector with the following representation:*

$$\hat{\beta}(\lambda) \asymp A(\Sigma, \lambda) \cdot \beta + B(\Sigma, \lambda) \cdot \sigma \cdot \frac{Z}{p^{1/2}}.$$

*Here $Z \sim \mathcal{N}(0, I_p)$ is a random vector that is stochastically dependent only on the noise $\varepsilon$, and $A, B$ are deterministic matrices defined by applying the scalar functions below to $\Sigma$:*

$$A(x, \lambda) = (c_p x + \lambda)^{-2}(c_p + c'_p)x, \qquad B(x, \lambda) = (c_p x + \lambda)^{-1}c_p x.$$

*Here $c_p := c(n, p, \Sigma, \lambda)$ is the unique positive solution of the fixed point equation*

$$1 - c_p = \frac{c_p}{n}\operatorname{tr}\left[\Sigma(c_p\Sigma + \lambda I)^{-1}\right]. \tag{2}$$

This result gives a precise representation of the ridge regression estimator. It is a sum of two terms: the true coefficient vector $\beta$ scaled by the matrix $A(\Sigma, \lambda)$, and the noise vector $Z$ scaled by the matrix $B(\Sigma, \lambda)$. The first term captures to what extent ridge regression recovers the "signal". Morever, the

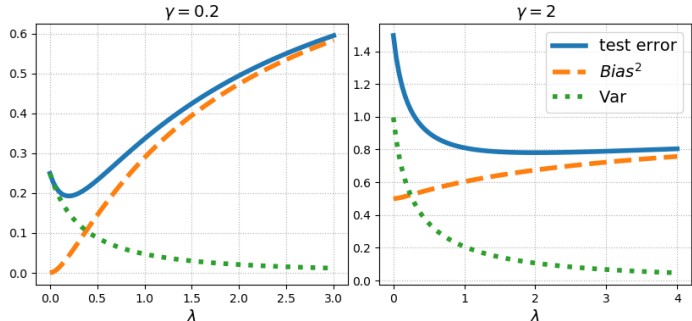

Figure 1: Ridge regression bias-variance tradeoff. Left: $\gamma = p/n = 0.2$; right: $\gamma = 2$. The data matrix $X$ has iid Gaussian entries. The coefficient $\beta$ has distribution $\beta \sim \mathcal{N}(0, I_p/p)$, while the noise $\varepsilon \sim \mathcal{N}(0, I_p)$.

noise term $Z$ is directly coupled with the noise in the original regression problem, and thus also the estimator. The result would not hold for an independent noise vector $Z$.

However, the coefficients are not fully explicit, as they depend on the unknown population covariance matrix $\Sigma$, as well as on the fixed-point variable $c_p$.

Some comments are in order:

1. **Structure of the proof.** The proof is quite non-elementary and relies on random matrix theory. Specifically, it uses the language of the recently developed "calculus of deterministic equivalents" (Dobriban & Sheng, 2018), and results by (Rubio & Mestre, 2011). A general takeaway is that for $n$ not much larger than $p$, the empirical covariance matrix $\widehat{\Sigma}$ is *not* a good estimator of the true covariance matrix $\Sigma$. However, the deviation of linear functionals of $\widehat{\Sigma}$, can be quantified. In particular, we have

$$(\widehat{\Sigma} + \lambda I)^{-1} \asymp (c_p \Sigma + \lambda I)^{-1},$$

   in the sense that linear combinations of the entries of the two matrices are close (see the proof for more details).

2. **Understanding the resolvent bias factor $c_p$.** Thus, $c_p$ can be viewed as a *resolvent bias factor*, which tells us by what factor $\Sigma$ is multiplied when evaluating the resolvent $(\widehat{\Sigma} + \lambda I)^{-1}$, and comparing it to its naive counterpart $(\Sigma + \lambda I)^{-1}$. It is known that $c_p$ is well defined, and this follows by a simple monotonicity argument, see Hachem et al. (2007); Rubio & Mestre (2011). Specifically, the left hand side of (2) is decreasing in $c_p$, while the right hand size is increasing in
   Also $c'_p$ is the derivative of $c_p$, when viewing it as a function of $z := -\lambda$. An explicit expression is provided in the proof in Section A.1, but is not crucial right now.

Here we discuss some implications of this representation.

**For uncorrelated features, $\Sigma = I_p$, $A$, $B$ reduce to multiplication by scalars.** Hence, each coordinate of the ridge regression estimator is simply a scalar multiple of the corresponding coordinate of $\beta$. One can use this to find the bias in each individual coordinate.

**Training error and optimal regularization parameter.** This theorem has implications for understanding the training error, and optimal regularization parameter of ridge regression. As it stands, the theorem itself only characterizes the behavior og *linear combinations* of the coordinates of the estimator. Thus, it can be directly applied to study the bias $\mathbb{E}\hat{\beta}(\lambda) - \beta$ of the estimator. However, it cannot directly be used to study the variance; as that would require understanding *quadratic functionals* of the estimator. This seems to require significant advances in random matrix theory, going beyond the results of Rubio & Mestre (2011). However, we show below that with additional assumptions on the structure of the parameter $\beta$, we can derive the MSE of the estimator in other ways.

We work in a *random-effects model*, where the $p$-dimensional regression parameter $\beta$ is random, each coefficient has zero mean $\mathbb{E}\beta_i = 0$, and is normalized so that $\text{Var}\beta_i = \alpha^2/p$. This ensures that the signal strength $\mathbb{E}\|\beta\|^2 = \alpha^2$ is fixed for any $p$. The asymptotically optimal $\lambda$ in this setting is always $\lambda^* = \gamma\sigma^2/\alpha^2$ see e.g., Tulino & Verdú (2004); Dicker (2016); Dobriban & Wager (2018). The ridge regression estimator with $\lambda = p\sigma^2/(n\alpha^2)$ is the posterior mean of $\beta$, when $\beta$ and $\varepsilon$ are normal random variables.

For a distribution $F$, we define the quantities

$$\theta_i(\lambda) = \int \frac{1}{(x+\lambda)^i} dF_\gamma(x),$$

($i = 1, 2, \ldots$). These are the moments of the resolvent and its derivatives (up to constants). We use the following loss functions: mean squared estimation error: $M(\hat{\beta}) = \mathbb{E}\|\hat{\beta} - \beta\|_2^2$, and residual or training error: $R(\hat{\beta}) = \mathbb{E}\left[\|\| Y - X\hat{\beta}\|_2^2\right]$.

**Theorem 2.2** (MSE and training error of ridge). *Suppose $\beta$ has iid entries with $\mathbb{E}\beta_i = 0$, $\text{Var}\,[\beta_i] = \alpha^2/p$, $i = 1, \ldots, p$ and $\beta$ is independent of $X$ and $\varepsilon$. Suppose $X$ is an arbitrary $n \times p$ matrix depending on $n$ and $p$, and the ESD of $X$ converges weakly to a deterministic distribution $F$ as $n, p \to \infty$ and $p/n \to \gamma$. Then the asymptotic MSE and residual error of the ridge regression estimator $\hat{\beta}(\lambda)$ has the form*

$$\lim_{n \to \infty} M(\hat{\beta}(\lambda)) = \alpha^2\lambda^2\theta_2 + \gamma\sigma^2[\theta_1 - \lambda\theta_2], \tag{3}$$

$$\lim_{n \to \infty} R(\hat{\beta}(\lambda)) = \alpha^2\lambda^2[\theta_1 - \lambda\theta_2] + \sigma^2\left[1 - \gamma(1 + \lambda\theta_1 - \lambda^2\theta_2)\right], \tag{4}$$

**Bias-variance tradeoff.** Building on this, we can also study the bias-variance tradeoff of ridge regression. Qualitatively, large $\lambda$ leads to more regularization, and decreases the variance. However, it also increases the bias. Our theory allows us to find the explicit formulas for the bias and variance as a function of $\lambda$. See Figure 1 for a plot and Sec. A.3 for the details. As far as we know, this is one of the few examples of high-dimensional asymptotic problems where the precise form of the bias and variance can be evaluated.

**Bias-variance tradeoff at optimal** $\lambda^* = \gamma\sigma^2/\alpha^2$**.** (see Figure 6) This can be viewed as the "pure" effect of dimensionality on the problem, keeping all other parameters fixed, and has intriguing properties. The variance first increases, then decreases with $\gamma$. In the "classical" low-dimensional case, most of the risk is due to variance, while in the "modern" high-dimensional case, most of it is due to bias. This is consistent with other phenomena in proportional-limit asymptotics, e.g., that the map between population and sample eigenvalue distributions is asymptotically deterministic (Marchenko & Pastur, 1967).

**Future applications.** This fundamental representation may have applications to important statistical inference questions. For instance, inference on the regression coefficient $\beta$ and the noise variance $\sigma^2$ are important and challenging problems. Can we use our representation to develop debiasing techniques for this task? This will be interesting to explore in future work.

## 3 CROSS-VALIDATION

How can we choose the regularization parameter? In practice, cross-validation (CV) is the most popular approach. However, it is well known that CV has a bias for estimating the error rate, because it uses a smaller number of samples than the full data size (e.g., Hastie et al., 2009, p. 243). In this section, we study related questions, proposing a bias-correction method for the optimal regularization parameter. This is closely connected to the previous section, because it relies on the same random-effects theoretical framework. In fact, our conclusions here are a direct consequence of the properties of that framework.

**Setup.** Suppose we split the $n$ datapoints (samples) into $K$ equal-sized subsets, each containing $n_0 = n/K$ samples. We use the $k$-th subset $(X_k, Y_k)$ as the validation set and the other $K - 1$ subsets $(X_{-k}, Y_{-k})$, with total sample size $n_1 = (K - 1)n/K$ as the training set. We find the ridge

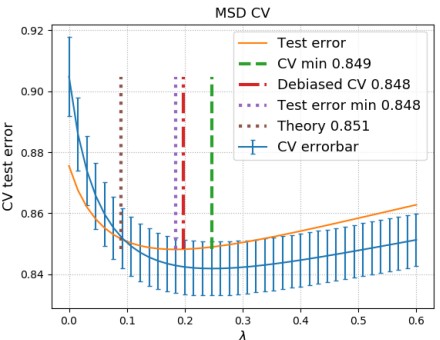 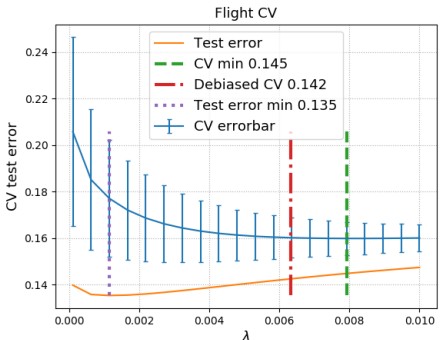

Figure 2: Left: Cross-validation on the Million Song Dataset (MSD, Bertin-Mahieux et al., 2011). For the error bar, we take $n = 1000$, $p = 90$, $K = 5$, and average over 90 different sub-datasets. For the test error, we train on 1000 training datapoints and fit on 9000 test datapoints. The debiased $\lambda$ reduces the test error by 0.00024, and the minimal test error is 0.8480. Right: Cross-validation on the flights dataset Wickham (2018). For the error bar, we take $n = 300$, $p = 21$, $K = 5$, and average over 180 different sub-datasets. For the test error, we train on 300 datapoints and fit on 27000 test datapoints. The debiased $\lambda$ reduces the test error by 0.0022, and the minimal test error is 0.1353.

regression estimator $\hat{\beta}_{-k}$, i.e.

$$\hat{\beta}_{-k}(\lambda) = \left( X_{-k}^\top X_{-k} + n_1 \lambda I_p \right)^{-1} X_{-k}^\top Y_{-k}.$$

The expected cross-validation error is, for isotropic covariance, i.e., $\Sigma = I$,

$$CV(\lambda) = \mathbb{E}\widehat{CV}(\lambda) = \mathbb{E}\left[ \frac{1}{K} \sum_{k=1}^K \|Y_k - X_k \hat{\beta}_{-k}(\lambda)\|_2^2 / n_0 \right] = \sigma^2 + \mathbb{E}\left[ \|\hat{\beta}_{-k} - \beta\|_2^2 \right].$$

**Bias in CV.** When $n, p$ tend to infinity so that $p/n \to \gamma > 0$, and in the random effects model with $\mathbb{E}\beta_i = 0$, $\mathrm{Var}\beta_i = \alpha^2/p$ described above, the minimizer of $CV(\lambda)$ tends to $\lambda_k^* = \tilde{\gamma}\sigma^2/\alpha^2$, where $\tilde{\gamma}$ is the limiting aspect ratio of $X_{-k}$, i.e. $\tilde{\gamma} = \gamma K/(K-1)$. Since the aspect ratios of $X_{-k}$ and $X$ differ, the limiting minimizer of the cross-validation estimator of the test error is biased for the limiting minimizer of the actual test error, which is $\lambda^* = \gamma\sigma^2/\alpha^2$.

**Bias-correction.** Suppose we have found $\hat{\lambda}_k^*$, the minimizer of $\widehat{CV}(\lambda)$. Afterwards, we usually refit ridge regression on the entire dataset, i.e., find

$$\hat{\beta}(\hat{\lambda}^*) = (X^\top X + \hat{\lambda}^* n I)^{-1} X^\top Y.$$

Based on our bias calculation, we propose to use a *bias-corrected* parameter

$$\hat{\lambda}^* := \hat{\lambda}_k^* \frac{K-1}{K}.$$

So if we use 5 folds, we should multiply the CV-optimal $\lambda$ by 0.8. We find it surprising that this theoretically justified bias-correction does not depend on any unknown parameters, such as $\beta, \alpha^2, \sigma^2$. While the bias of CV is widely known, we are not aware that this bias-correction for the regularization parameter has been proposed before.

**Numerical examples.** Figure 2 shows on two empirical data examples that the debiased estimator gets closer to the optimal $\lambda$ than the original minimizer of the CV. However, in this case it does not significantly improve the test error. Simulation results in Section A.4 also show that the bias-correction correctly shrinks the regularization parameter and decreases the test error. We also consider examples where $p \gg n$ (i.e., $\gamma \gg 1$), because this is a setting where it is known that the bias of CV can be large (Tibshirani & Tibshirani, 2009). However, in this case, we do not see a significant improvement.

**Extensions.** The same bias-correction idea also applies to train-test validation. In addition, there is a special fast "short-cut" for leave-one-out cross-validation in ridge regression (e.g., Hastie et al.,

2009, p. 243), which has the same cost as one ridge regression. The minimizer converges to $\lambda^*$ (Hastie et al., 2019). However, we think that the bias-correction idea is still valuable, as the idea applies beyond ridge regression: CV selects regularization parameters that are too large. See Section A.5 for more details and experiments comparing different ways of choosing the regularization parameter.

## 4 SKETCHING

A final important question about ridge regression is how to compute it in practice. In this section, we study that problem *in the same high-dimensional model* used throughout our paper. The computation complexity of ridge regression, $O(np \min(n, p))$, can be intractable in modern large-scale data analysis. Sketching is a popular approach to reducing the time complexity by reducing the sample size and/or dimension, usually by random projection or sampling (e.g. Mahoney, 2011; Woodruff, 2014; Drineas & Mahoney, 2016). Specifically, *primal sketching* approximates the sample covariance matrix $X^\top X/n$ by $X^\top L^\top L X/n$, where $L$ is an $m \times n$ sketching matrix, and $m < n$. If $L$ is chosen as a suitable random matrix, then this can still approximate the original sample covariance matrix. Then the primal sketched ridge regression estimator is

$$\hat{\beta}_p = \left( X^\top L^\top L X/n + \lambda I_p \right)^{-1} X^\top Y/n. \tag{5}$$

*Dual sketching* reduces $p$ instead. An equivalent expression for ridge regression is $\hat{\beta} = n^{-1} X^\top \left( XX^\top/n + \lambda I_n \right)^{-1} Y$. Dual sketched ridge regression reduces the computation cost of the Gram matrix $XX^\top$, approximating it by $XRR^\top X^\top$ for another sketching matrix $R \in \mathbb{R}^{p \times d}$ ($d < p$), so

$$\hat{\beta}_d = X^\top \left( XRR^\top X^\top/n + \lambda I_n \right)^{-1} Y/n. \tag{6}$$

The sketching matrices $R$ and $L$ are usually chosen as random matrices with iid entries (e.g., Gaussian ones) or as orthogonal matrices. In this section, we study the asymptotic MSE for both orthogonal (Section 4.1) and Gaussian sketching (Section 4.2). We also mention *full sketching*, which performs ridge after projecting down both $X$ and $Y$. In section A.11, we find its MSE. However, the other two methods have better tradeoffs, and we can empirically get better results for the same computational cost.

### 4.1 ORTHOGONAL SKETCHING

First we consider primal sketching with orthogonal projections. These can be implemented by subsampling, Haar distributed matrices, or subsampled randomized Hadamard transforms (Sarlos, 2006). We recall that the standard *Marchenko-Pastur (MP) law* is the probability distribution which is the limit of the ESD of $X^\top X/n$, when the $n \times p$ matrix $X$ has iid standard Gaussian entries, and $n, p \to \infty$ so that $p/n \to \gamma > 0$, which has an explicit density (Marchenko & Pastur, 1967; Bai & Silverstein, 2010).

**Theorem 4.1** (Primal orthogonal sketching). *Suppose $\beta$ has iid entries with $\mathbb{E}\beta_i = 0$, $\text{Var}[\beta_i] = \alpha^2/p$, $i = 1, \ldots, p$ and $\beta$ is independent of $X$ and $\varepsilon$. Suppose $X$ has iid standard normal entries.*

*We compute primal sketched ridge regression (5) with an $m \times n$ orthogonal matrix $L$ ($m < n$, $LL^\top = I_m$). Let $n, p$ and $m$ tend to infinity with $p/n \to \gamma \in (0, \infty)$ and $m/n \to \xi \in (0, 1)$. Then the MSE of $\hat{\beta}_p(\lambda)$ has the limit*

$$M(\lambda) = \alpha^2 \frac{\left[ (\lambda + \xi - 1)^2 + \gamma(1 - \xi) \right] \theta_2 \left( \frac{\gamma}{\xi}, \frac{\lambda}{\xi} \right)}{\xi^2} + \gamma \sigma^2 \frac{\xi \theta_1 \left( \frac{\gamma}{\xi}, \frac{\lambda}{\xi} \right) - (\lambda + \xi - 1)\theta_2 \left( \frac{\gamma}{\xi}, \frac{\lambda}{\xi} \right)}{\xi^2}, \tag{7}$$

*where $\theta_i(\gamma, \lambda) = \int (x + \lambda)^{-i} dF_\gamma(x)$ and $F_\gamma$ is the standard Marchenko-Pastur law with aspect ratio $\gamma$.*

**Structure of the proof.** The proof is in Section A.6, with explicit formulas in Section A.6.1. The $\theta_i$ are related to the resolvent of the MP law and its derivatives. In the proof, we decompose the

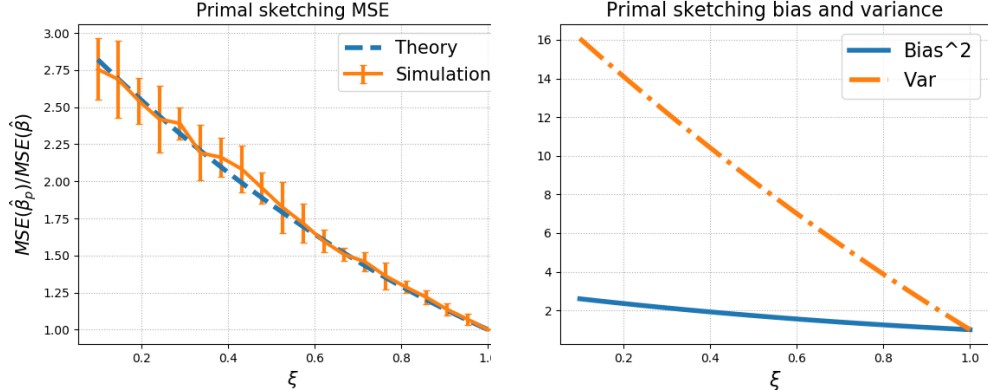

Figure 3: Primal orthogonal sketching with $n = 500$, $\gamma = 5$, $\lambda = 1.5$, $\alpha = 3$, $\sigma = 1$. Left: MSE of primal sketching normalized by the MSE of ridge regression. The error bar is the standard deviation over 10 repetitions. Right: Bias and variance of primal sketching normalized by the bias and variance of ridge regression, respectively.

MSE as the sum of variance and squared bias, both of which further reduce to the traces of certain random matrices, whose limits are determined by the MP law $F_\gamma$ and $\lambda$. The two terms on the RHS of Equation (7) are the limits of squared bias and variance, respectively. There is an additional key step in the proof, which introduces *the orthogonal complement $L_1$* of the matrix $L$ such that $L^\top L + L_1^\top L_1 = I_n$, which leads to some Gaussian random variables appearing in the proof, and simplifies calculations.

**Simulations.** A simulation in Figure 3 (left) shows a good match with our theory. It also shows that sketching does not increase the MSE too much. In this case, by reducing the sample size to half the original one, we only increase the MSE by a factor of 1.05. This shows sketching can be very effective. We also see in Figure 3 (right) that variance is compromised much more than bias.

**Robustness to tuning parameter.** The reader may wonder how strongly this depends on the choice of the regularization parameter $\lambda$. Perhaps ridge regression works poorly with this $\lambda$, so sketching cannot worsen it too much? What happens if we take the optimal $\lambda$ instead of a fixed one? In experiments in Section A.12 we show that the behavior is quite robust to the choice of regularization parameter.

The next theorem states a result for dual orthogonal sketching.

**Theorem 4.2** (Dual orthogonal sketching). *Under the conditions of Theorem 4.1, we compute the dual sketched ridge regression with an orthogonal $p \times d$ sketching matrix $R$ ($d \leqslant p$, $R^\top R = I_d$). Let $n, p$ and $d$ go to infinity with $p/n \to \gamma \in (0, \infty)$ and $d/n \to \zeta \in (0, \gamma)$. Then the MSE of $\hat{\beta}_d(\lambda)$ has the limit*

$$\frac{\alpha^2}{\gamma} \left[ \gamma - 1 + (\lambda - \gamma + \zeta)^2 \bar{\theta}_2(\zeta, \lambda) + (\gamma - \zeta) \bar{\theta}_1^2(\zeta, \lambda) \right] + \sigma^2 \left[ \bar{\theta}_1(\zeta, \lambda) - (\lambda + \zeta - \gamma) \bar{\theta}_2(\zeta, \lambda) \right],$$

*where $\bar{\theta}_i(\zeta, \lambda) = (1 - \zeta)/\lambda^i + \zeta \int (x + \lambda)^{-i} dF_\zeta(x)$, and $F_\zeta$ is the standard Marchenko-Pastur law.*

**Proof structure and simulations.** The proof in Section A.7 follows similar path to the previous one. Here $\bar{\theta}_i$ comes in because of the companion Stieltjes transform of MP law. The simulation results shown in Figure 11 agrees well with our theory. They are similar to the ones before: sketching has favorable properties, and the bias increases less than the variance.

**Optimal tuning parameters.** For both primal and dual sketching, the optimal regularization parameter minimizing the MSE seems analytically intractable. Instead, we use a numerical approach in our experiments, based on a binary search. Since this is one-dimensional problem, there are no numerical issues. See Figure 13 in Section A.12.3.

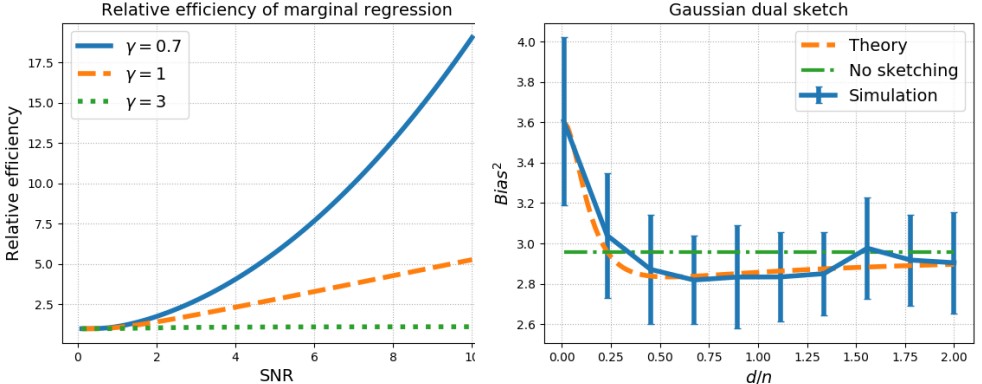

Figure 4: Left: Ratio of optimal MSE of marginal regression to that of optimally tuned ridge regression, for three values of $\gamma = p/n$, as a function of the SNR $\alpha^2/\sigma^2$. Right: Gaussian dual sketch when there is no noise. $\gamma = 0.4$, $\alpha = 1$, $\lambda = 1$ (both for original and sketching). Standard error over 50 experiments.

### 4.1.1 EXTREME PROJECTION — MARGINAL REGRESSION

It is of special interest to investigate *extreme projections*, where the sketching dimension is much reduced compared to the sample size, so $m \ll n$. This corresponds to $\xi = 0$. This can also be viewed as a scaled *marginal regression* estimator, i.e., $\hat{\beta} \propto X^\top Y$. For dual sketching, the same case can be recovered with $\zeta = 0$. Another interest of studying this special case is that the formula for MSE simplifies a lot.

**Theorem 4.3** (Marginal regression). *Under the same assumption as Theorem 4.1, let $\xi = 0$. Then the form of the MSE is $M(\lambda) = \left[\alpha^2 \left[(\lambda - 1)^2 + \gamma\right] + \sigma^2 \gamma\right]/\lambda^2$. Moreover, the optimal $\lambda^*$ that minimizes this equals $\gamma \sigma^2/\alpha^2 + 1 + \gamma$ and the optimal MSE is $M(\lambda^*) = \alpha^2 \left(1 - \alpha^2/[\alpha^2(1 + \gamma) + \gamma \sigma^2]\right)$.*

The proof is in Section A.8. When is the optimal MSE of marginal regression small? Compared to the MSE of the zero estimator $\alpha^2$, it is small when $\gamma(\sigma^2/\alpha^2 + 1) + 1$ is large. In Figure 4 (left), we compare marginal and ridge regression for different aspect ratios and SNR. When the signal to noise ratio (SNR) $\alpha^2/\sigma^2$ is small or the aspect ratio $\gamma$ is large, marginal regression does not increase the MSE much. As a concrete example, if we take $\alpha^2 = \sigma^2 = 1$ and $\gamma = 0.7$, the marginal MSE is $1 - 1/2.4 \approx 0.58$. The optimal ridge MSE is about $0.52$, so their ratio is only ca. $0.58/0.52 \approx 1.1$. It seems quite surprising that a simple-minded method like marginal regression can work so well. However, the reason is that when the SNR is small, we cannot expect ridge regression to have good performance. Large $\gamma$ can also be interpreted as small SNR, where ridge regression works poorly and sketching does not harm performance too much.

### 4.2 GAUSSIAN SKETCHING

In this section, we study Gaussian sketching. The following theorem states the bias of dual Gaussian sketching. The bias is enough to characterize the performance in the high SNR regime where $\alpha/\sigma \to \infty$, and we discuss the extension to low SNR after the proof.

**Theorem 4.4** (Bias of dual Gaussian sketch). *Suppose $X$ is an $n \times p$ standard Gaussian random matrix. Suppose also that $R$ is a $p \times d$ matrix with i.i.d. $\mathcal{N}(0, 1/d)$ entries. Then the bias of dual sketch has the expression $\text{Bias}^2(\hat{\beta}_d) = \alpha^2 + \alpha^2/\gamma \cdot [m'(z) - 2m(z)]|_{z=0}$, where $m$ is a function described below, and $m'(z)$ denotes the derivative of $m$ w.r.t. $z$. Below, we use the branch of the square root with positive imaginary part.*

*The function $m$ is characterized by its inverse function, which has the explicit formula $m^{-1}(z) = 1/[1 + z/\zeta] - [\gamma + 1 - \sqrt{(\gamma - 1)^2 + 4\lambda z}]/(2z)$ for complex $z$ with positive imaginary part.*

**About the proof.** The proof is in Section A.9.We mention that the same result holds when the matrices involved have iid non-Gaussian entries, but the proof is more technical. The current proof is based on free probability theory (e.g., Voiculescu et al., 1992; Hiai & Petz, 2006; Couillet & Debbah, 2011). The function $m$ is the Stieltjes transform of the free additive convolution of a standard MP law $F_{1/\xi}$ and a scaled inverse MP law $\lambda/\gamma \cdot F_{1/\gamma}^{-1}$ (see the proof).

**Numerics.** To evaluate the formula, we note that $m^{-1}(m(0)) = 0$, so $m(0)$ is a root of $m^{-1}$. Also, $dm(0)/dz$ equals $1/(dm^{-1}(y)/dy|_{y=m(0)})$, the reciprocal of the derivative of $m^{-1}$ evaluated at $m(0)$. We use binary search to find the numerical solution. The theoretical result agrees with the simulation quite well, see Figure 4.

Somewhat unexpectedly, the MSE of dual sketching can be below the MSE of ridge regression, see Figure 4. This can happen when the original regularization parameter is suboptimal. As $d$ grows, the MSE of Gaussian dual sketching converges to that of ridge regression.

We have also found the bias of primal Gaussian sketching. However, stating the result requires free probability theory, and so we present it in the Appendix, see Theorem A.1. To further validate our results, we present additional simulations in Sec. A.12, for both fixed and optimal regularization parameters after sketching. A detailed study of the computational cost for sketching in Sec. A.13 concludes, as expected, that primal sketching can reduce cost when $p < n$, while dual sketching can reduce it when $p > n$; and also provides a more detailed analysis.

ACKNOWLEDGMENTS

The authors thank Ken Clarkson for helpful discussions and for providing the reference Chen et al. (2015). ED was partially supported by NSF BIGDATA grant IIS 1837992. SL was partially supported by a Tsinghua University Summer Research award. A version of our manuscript is available on arxiv at `https://arxiv.org/abs/1910.02373`.

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

# A  APPENDIX

## A.1  PROOF OF THEOREM 2.1

If $p/n \to \gamma$ and the spectral distribution of $\Sigma$ converges to $H$, we have by the general Marchenko-Pastur (MP) theorem of Rubio and Mestre (Rubio & Mestre, 2011), that

$$(\widehat{\Sigma} + \lambda I)^{-1} \asymp (c_p \Sigma + \lambda I)^{-1},$$

where $c_p := c(n, p, \Sigma, \lambda)$ is the unique positive solution of the fixed point equation

$$1 - c_p = \frac{c_p}{n} \operatorname{tr}\left[\Sigma(c_p \Sigma + \lambda I)^{-1}\right].$$

Here, using the terminology of the calculus of deterministic equivalents (Dobriban & Sheng, 2018), two sequences of (not necessarily symmetric) $n \times n$ matrices $A_n, B_n$ of growing dimensions are *equivalent*, and we write

$$A_n \asymp B_n$$

if $\lim_{n \to \infty} \operatorname{tr}\left[C_n(A_n - B_n)\right] = 0$ almost surely, for any sequence $C_n$ of (not necessarily symmetric) $n \times n$ deterministic matrices with bounded trace norm, i.e., such that $\limsup \|C_n\|_{tr} < \infty$ (Dobriban & Sheng, 2018). Informally, linear combinations of the entries of $A_n$ can be approximated by the entries of $B_n$.

We start with

$$\hat{\beta} = \left(X^\top X/n + \lambda I_p\right)^{-1} X^\top Y/n = \left(X^\top X/n + \lambda I_p\right)^{-1} \frac{X^\top(X\beta + \varepsilon)}{n}$$

$$= (\widehat{\Sigma} + \lambda I_p)^{-1}\widehat{\Sigma}\beta + (\widehat{\Sigma} + \lambda I_p)^{-1}\frac{X^\top \varepsilon}{n}.$$

Then, by the general MP law written in the language of the calculus of deterministic equivalents

$$(\widehat{\Sigma} + \lambda I_p)^{-1}\widehat{\Sigma} = I_p - \lambda(\widehat{\Sigma} + \lambda I_p)^{-1} \asymp I_p - \lambda(c_p \Sigma + \lambda I)^{-1} = c_p \Sigma(c_p \Sigma + \lambda I)^{-1}.$$

By the definition of equivalence for vectors,

$$(\widehat{\Sigma} + \lambda I_p)^{-1}\widehat{\Sigma}\beta \asymp c_p \Sigma(c_p \Sigma + \lambda I)^{-1}\beta.$$

We note a subtle point here. The rank of the matrix $M := (\widehat{\Sigma} + \lambda I_p)^{-1}\widehat{\Sigma}$ is at most $n$, and so it is not a full rank matrix when $n < p$. In contrast, $c_p \Sigma(c_p \Sigma + \lambda I)^{-1}$ can be a full rank matrix. Therefore, for the vectors $\beta$ in the null space of $\widehat{\Sigma}$, which is also the null space of $X$, we certainly have that the two sides are not equal. However, here we assumed that the matrix $X$ is random, and so its null space is a random $\max(p-n, 0)$ dimensional linear space. Therefore, for any fixed vector $\beta$, the random matrix $M$ will not contain it in its null space with high probability, and so there is no contradiction.

We should also derive an asymptotic equivalent for

$$(\widehat{\Sigma} + \lambda I_p)^{-1}\frac{X^\top \varepsilon}{n}.$$

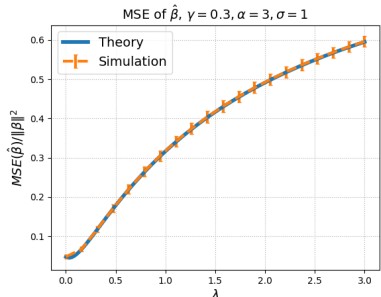

Figure 5: Simulation for ridge regression. We take $n = 1000$, $\lambda = 0.3$. Also, $X$ has iid $\mathcal{N}(0, 1)$ entries, $\beta_i \sim_{iid} \mathcal{N}(0, \alpha^2/p)$, $\varepsilon_i \sim_{iid} \mathcal{N}(0, \sigma^2)$, with $\alpha = 3, \sigma = 1$. The standard deviations are over 50 repetitions. The theoretical lines are plotted according to Theorem 2.2. The MSE is normalized by the norm of $\beta$.

Suppose we have Gaussian noise, and let $Z \sim \mathcal{N}(0, I_p)$. Then we can write

$$(\widehat{\Sigma} + \lambda I_p)^{-1}\frac{X^\top \varepsilon}{n} =_d (\widehat{\Sigma} + \lambda I_p)^{-1}\widehat{\Sigma}^{1/2}\frac{\sigma Z}{n^{1/2}}.$$

So the question reduces to finding a deterministic equivalent for $h(\widehat{\Sigma})$, where $h(x) = (x + \lambda)^{-2}x$. Note that

$$h(x) = (x + \lambda)^{-2}x = (x + \lambda)^{-2}(x + \lambda - \lambda) = (x + \lambda)^{-1} - \lambda(x + \lambda)^{-2}.$$

By the calculus of deterministic equivalents: $(\widehat{\Sigma} + \lambda)^{-1} \asymp (c_p\Sigma + \lambda I)^{-1}$. Moreover, fortunately the limit of the second part was recently calculated in (Dobriban & Sheng, 2019). This used the so-called "differentiation rule" of the calculus of deterministic equivalents to find

$$(\widehat{\Sigma} + \lambda)^{-2} \asymp (c_p\Sigma + \lambda I)^{-2}(I - c_p'\Sigma).$$

The derivative $c_p' = dc_p/dz$ has been found in Dobriban & Sheng (2019), in the proof of Theorem 3.1, part 2b. The result is (with $\gamma_p = p/n$, $H_p$ the spectral distribution of $\Sigma$, and $T$ a random variable distributed according to $H_p$)

$$c_p' = \frac{\gamma_p \mathbb{E}_{H_p}\frac{c_p T}{(c_p T - z)^2}}{-1 + \gamma_p z \mathbb{E}_{H_p}\frac{T}{(c_p T - z)^2}}. \tag{8}$$

So, we find the final answer

$$(\widehat{\Sigma} + \lambda I_p)^{-1}\widehat{\Sigma}^{1/2} \asymp A(\Sigma, \lambda) := (c_p\Sigma + \lambda I)^{-1} - \lambda(c_p\Sigma + \lambda I)^{-2}(I - c_p'\Sigma).$$

## A.2 RISK ANALYSIS

Figure 5 shows a simulation result. We see a good match between theory and simulation.

### A.2.1 PROOF OF THEOREM 2.2

*Proof.* The MSE of $\hat{\beta}$ has the form

$$\mathbb{E}\|\hat{\beta} - \beta\|^2 = \text{bias}^2 + \delta^2,$$

where

$$\text{bias}^2 = \mathbb{E}\left[\left\|(X^\top X/n + \lambda I_p)^{-1}X^\top X/n\beta - \beta\right\|_2^2\right],$$

$$\delta^2 = \sigma^2 \mathbb{E}\left[\left\|(X^\top X/n + \lambda I_p)^{-1}n^{-1}X^\top\right\|_F^2\right].$$

We assume that $X$ has iid entries of zero mean and unit variance, and that $\mathbb{E}\beta = 0$, $\mathrm{Var}\,[\beta] = \alpha^2/pI_p$. As $p/n \to \gamma$ as $n$ goes to infinity, the ESD of $\frac{1}{n}X^\top X$ converges to the MP law $F_\gamma$. So we have

$$\mathrm{bias}^2 = \mathbb{E}\left[\left\|\lambda\left(X^\top X/n + \lambda I_p\right)^{-1}\beta\right\|_2^2\right]$$
$$= \alpha^2\lambda^2\mathbb{E}\left[\frac{1}{p}\,\mathrm{tr}[(X^\top X/n + \lambda I_p)^{-2}]\right] \to \alpha^2\lambda^2\int\frac{1}{(x+\lambda)^2}dF_\gamma(x),$$

and

$$\delta^2 = \frac{\sigma^2}{n^2}\mathbb{E}\left[\mathrm{tr}[(X^\top X/n + \lambda I_p)^{-2}X^\top X]\right]$$
$$= \frac{\sigma^2}{n}\mathbb{E}\left[\mathrm{tr}[(X^\top X/n + \lambda I_p)^{-1} - \lambda(X^\top X/n + \lambda I_p)^{-2}]\right]$$
$$\to \sigma^2\gamma\left[\int\frac{1}{x+\lambda}dF_\gamma(x) - \lambda\int\frac{1}{(x+\lambda)^2}dF_\gamma(x)\right].$$

Denoting $\theta_i(\gamma,\lambda) = \int\frac{1}{(x+\lambda)^i}dF_\gamma(x)$, then

$$AMSE(\hat{\beta}) = \alpha^2\lambda^2\theta_2 + \gamma\sigma^2[\theta_1 - \lambda\theta_2]. \qquad (9)$$

For the standard Marchenko-Pastur law (i.e., when $\Sigma = I_p$), we have the explicit forms of $\theta_1$ and $\theta_2$. Specifically,

$$\theta_1 = \int\frac{1}{x+\lambda}dF_\gamma(x) = -\frac{1}{2}\left[\frac{2(1+\lambda)}{\lambda\gamma} + \frac{2}{\sqrt{\gamma}\lambda}z_2\right]$$

where

$$z_2 = -\frac{1}{2}\left[(\sqrt{\gamma} + \frac{1+\lambda}{\sqrt{\gamma}}) + \sqrt{(\sqrt{\gamma} + \frac{1+\lambda}{\sqrt{\gamma}})^2 - 4}\right].$$

It is known that the limiting Stieltjes transform $m_{F_\gamma} := m_\gamma$ of $\widehat{\Sigma}$ has the explicit form (Marchenko & Pastur, 1967):

$$m_\gamma(z) = \frac{(z+\gamma-1) + \sqrt{(z+\gamma-1)^2 - 4z\gamma}}{-2z\gamma}.$$

As usual in the area, we use the principal branch of the square root of complex numbers. Hence $\theta_1 = \frac{(-\lambda+\gamma-1)+\sqrt{(-\lambda+\gamma-1)^2+4\lambda\gamma}}{2\lambda\gamma}$. Also

$$\theta_2(\gamma,\lambda) = \int\frac{1}{(x+\lambda)^2}dF_\gamma(x) = -\int\frac{d}{d\lambda}\frac{1}{x+\lambda}dF_\gamma(x)$$
$$= -\frac{d}{d\lambda}\theta_1 = -\frac{1}{\gamma\lambda^2} + \frac{1}{\sqrt{\gamma}}\frac{d}{d\lambda}\frac{z_2}{\lambda}$$
$$= -\frac{1}{\gamma\lambda^2} + \frac{\gamma+1}{2\gamma\lambda^2} - \frac{1}{2\sqrt{\gamma}}\left[\frac{\lambda+\gamma+1}{\gamma\lambda\sqrt{(\sqrt{\gamma}+\frac{1+\lambda}{\sqrt{\gamma}})^2-4}} - \frac{\sqrt{(\sqrt{\gamma}+\frac{1+\lambda}{\sqrt{\gamma}})^2-4}}{\lambda^2}\right]$$

For the residual,

$$\mathbb{E}\left[\frac{1}{n}\|Y - X\hat{\beta}\|_2^2|X\right] = \alpha^2\lambda^2\frac{1}{p}\,\mathrm{tr}[(X^\top X/n + \lambda I_p)^{-1} - \lambda(X^\top X/n + \lambda I_p)^{-2}]$$
$$+ \sigma^2\frac{1}{n}[\mathrm{tr}(I_n) - 2\,\mathrm{tr}\left(X^\top X/n + \lambda I_p\right)^{-1}X^\top X/n + \mathrm{tr}\left(\left(X^\top X/n + \lambda I_p\right)^{-1}X^\top X/n\right)^2].$$

Next,

$$\mathbb{E}\left[\frac{1}{p}\,\mathrm{tr}\big[\big(\big(X^\top X/n + \lambda I_p\big)^{-1} X^\top X/n\big)^2\big]\right] = \mathbb{E}\left[\frac{1}{p}\,\mathrm{tr}\big[\big(I_p - \lambda\,\big(X^\top X/n + \lambda I_p\big)^{-1}\big)^2\big]\right]$$
$$\rightarrow 1 - 2\lambda\theta_1 + \lambda^2\theta_2.$$

Therefore

$$\mathbb{E}\left[\frac{1}{n}\|Y - X\hat\beta\|_2^2\right] \rightarrow \alpha^2\lambda^2[\theta_1 - \lambda\theta_2] + \sigma^2\left[1 - 2\gamma(1 - \lambda\theta_1) + \gamma(1 - 2\lambda\theta_1 + \lambda^2\theta_2)\right]$$
$$= \alpha^2\lambda^2[\theta_1 - \lambda\theta_2] + \sigma^2\left[1 - \gamma(1 + \lambda\theta_1 - \lambda^2\theta_2)\right].$$

$\square$

### A.3 BIAS-VARIANCE TRADEOFF

The limiting MSE decomposes into a limiting squared bias and variance. The specific forms of these are

$$\mathrm{bias}^2 = \alpha^2 \int \frac{\lambda^2}{(x+\lambda)^2}\,dF_\gamma(x), \qquad \mathrm{var} = \gamma\sigma^2 \int \frac{x}{(x+\lambda)^2}\,dF_\gamma(x).$$

See Figure 1 for a plot. We can make several observations.

1. The bias increases with $\lambda$, starting out at zero for $\lambda = 0$ (linear regression), and increasing to $\alpha^2$ as $\lambda \to \infty$ (zero estimator).
2. The variance decreases with $\lambda$, from $\gamma\sigma^2 \int x^{-1} dF_\gamma(x)$ to zero.
3. In the setting plotted in the figure, when $\alpha^2$ and $\sigma^2$ are roughly comparable, there are additional qualitative properties we can investigate. When $\gamma$ is small, the regularization parameter $\lambda$ influences the bias more strongly than the variance (i.e., the derivative of the normalized quantities in the range plotted is generally larger for the normalized squared bias). In contrast when $\gamma$ is large, the variance is influenced more.

Next we consider how bias and variance change with $\gamma$ at the optimal $\lambda^* = \gamma\sigma^2/\alpha^2$. This can be viewed as the "pure" effects of dimensionality on the problem, keeping all other parameters fixed. Ineed, $\alpha^2/\sigma^2$ can be viewed as the signal-to-noise ratio (SNR), and is fixed. This analysis allows us to study for the best possible estimator (ridge regression, a Bayes estimator), behaves with the dimension. We refer to Figure 6, where we make some specific choices of $\alpha$ and $\sigma$.

1. Clearly the overall risk increases, as the problem becomes harder with increasing dimension. This is in line with our intuition.
2. The classical bias-variance tradeoff can be summarized by the equation

$$\mathrm{bias}^2(\lambda) + \mathrm{var}(\lambda) \geqslant M^*(\alpha, \gamma),$$

   where we made explicit the dependence of the bias and variance on $\lambda$, and where $M^*(\alpha, \gamma)$ is the minimum MSE achievable, also known as the Bayes error, for which there are explicit formulas available (Tulino & Verdú, 2004; Dobriban & Wager, 2018).
3. The variance first increases, then decreases with $\gamma$. This shows that in the "classical" low-dimensional case, most of the risk is due to variance, while in the "modern" high-dimensional case, most of it is due to bias. This observation is consistent with other phenomena in proportional-limit asymptotics, for instance that the map between population and sample eigenvalue distributions is asymptotically deterministic (Marchenko & Pastur, 1967; Bai & Silverstein, 2010).

### A.4 SIMULATIONS WITH CROSS-VALIDATION

See Figure 7. We consider both small and large $\gamma$. Our bias-correction procedure shrinks the $\lambda$ to the correct direction and decreases the test error. It is also shown that the one-standard-error rule (e.g., Hastie et al., 2009) does not perform well here.

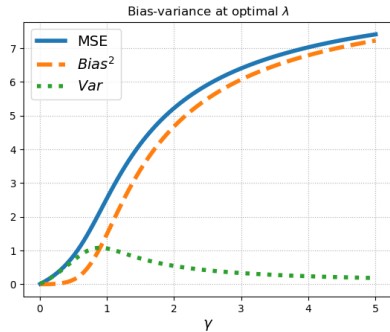

Figure 6: Bias-variance tradeoff at optimal $\lambda^* = \gamma \sigma^2 / \alpha^2$, when $\alpha = 3, \sigma = 1$.

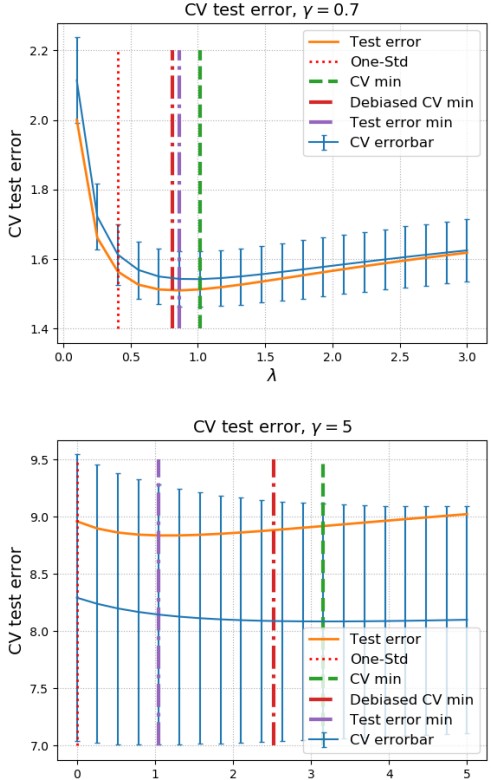

Figure 7: Left: we generate a training set ($n = 1000, p = 700, \gamma = 0.7, \alpha = \sigma = 1$) and a test set ($n_{test} = 500$) from the same distribution. We split the training set into $K = 5$ equally sized folds and do cross-validation. The blue error bars plot the mean and standard error of the $K$ test errors. The red dotted line indicates the "one-standard-error" location. The green dashed line indicates the optimal $\lambda^*_{CV}$ obtained by $k$-fold cross-validation, while the red dashed-dotted line indicates the debiased version $\frac{K-1}{K}\lambda^*_{CV}$. The orange line plots the test error when training on the whole training set and fit on the whole test set, and the purple dashed-dotted line indicates the minimal $\lambda^*_{test}$. The test error is 1.513 at $\lambda^*_{CV}$ and 1.510 at $\frac{K-1}{K}\lambda^*_{CV}$. So the bias-correction decreases the test error by about 0.003. Right: we take $n = 200, p = 1000, \gamma = 5, \alpha = 3, \sigma = 1$. The bias-correction decreases the test error from 8.92 to 8.89, so it decreases by 0.03.

A.5    CHOOSING THE REGULARIZATION PARAMETER- ADDITIONAL DETAILS

Another possible prediction method is to use the average of the ridge estimators computed during cross-validation. Here it is also natural to use the CV-optimal regularization parameters, averaging $\hat{\beta}_{-k}(\hat{\lambda}_k^*)$, i.e.

$$\hat{\beta}_{avg}(\hat{\lambda}_k^*) = \frac{1}{K} \sum_{k=1}^{K} \hat{\beta}_{-k}(\hat{\lambda}_k^*).$$

This has the advantage that it does not require refitting the ridge regression estimator, and also that we use the optimal regularization parameter.

A.5.1    TRAIN-TEST VALIDATION

The same bias in the regularization parameter also applies to train-test validation. Since the number of samples is changed when restricting to the training set, the optimal $\lambda$ chosen by train-test validation is also biased for the true regularization parameter minimizing the test error. We will later see in simulations (Figure 8) that retraining the ridge regression estimator on the whole data will still significantly improve the performance (this is expected based on our results on CV). For prediction, here we can also use ridge regression on the training set. This effectively reduces sample size $n \to n_{train}$, where $n_{train}$ is the sample size of the training set. However, if the training set grows such that $n/n_{train} \to 1$ while $n_{train} \to \infty$, the train-test split has asymptotically optimal performance.

A.5.2    LEAVE-ONE-OUT

There is a special "short-cut" for leave-one-out in ridge regression, which saves us from burdensome computation. Write $loo(\lambda)$ for the leave-one-out estimator of prediction error with parameter $\lambda$. Instead of doing ridge regression $n$ times, we can calculate the error explicitly as

$$loo(\lambda) = \frac{1}{n} \sum_{i=1}^{n} \left[ \frac{Y_i - X_i^\top \hat{\beta}(\lambda)}{1 - S_{ii}(\lambda)} \right]^2.$$

where $S(\lambda) = X(X^\top X + n\lambda I)^{-1} X^\top$. The minimizer of $loo(\lambda)$ is asymptotically optimal, i.e., it converges to $\lambda^*$ (Hastie et al., 2019). However, the computational cost of this shortcut is the same as that of a train-test split. Therefore, the method described above has the same asymptotic performance.

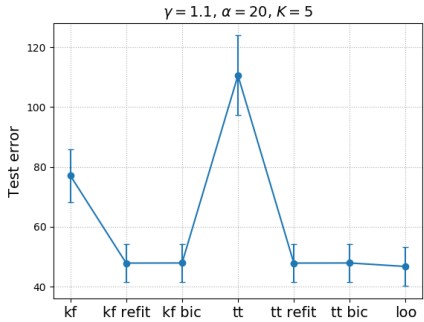

Figure 8: Comparing different ways of doing cross-validation. We take $n = 500$, $p = 550$, $\alpha = 20$, $\sigma = 1$, $K = 5$. As for train-test validation, we take $80\%$ of samples to be training set and the rest $20\%$ be test set. The error bars are the mean and standard deviation over 20 repetitions.

**Simulations:** Figure 8 shows simulation results comparing different cross-validation methods:

1. kf — k-fold cross-validation by taking the average of the ridge estimators at the CV-optimal regularization parameter.

2. kf refit — k-fold cross-validation by refitting ridge regression on the whole dataset using the CV-optimal regularization parameter.
3. kf bic — k-fold cross-validation by refitting ridge regression on the whole dataset using the CV-optimal regularization parameter, with bias correction.
4. tt — train-test validation, by using the ridge estimator computed on the train data, at the validation-optimal regularization parameter. Note: we expect this to be similar, but worse than the "kf" estimator.
5. tt refit — train-test validation by refitting ridge regression on the whole dataset, using the validation-optimal regularization parameter. Note: we expect this to be similar, but slightly worse than the "kf refit" estimator.
6. tt bic — train-test validation by refitting ridge regression on the whole dataset using the CV-optimal regularization parameter, with bias correction.
7. loo — leave-one-out

Figure 8 shows that the naive estimators (kf and tt) can be quite inaccurate without refitting or bias correction. However, if we either refit or bias-correct, the accuracy improves. In this case, there seems to be no significant difference between the various methods.

### A.6   PROOF OF THEOREM 4.1

*Proof.* Suppose $m/n \to \xi$ as $n$ goes to infinity. For $\hat{\beta}_p$, we have

$$\text{bias}^2 = \mathbb{E}\left[\left\|\left(X^\top L^\top L X/n + \lambda I_p\right)^{-1} X^\top X/n\beta - \beta\right\|_2^2\right],$$

$$\delta^2 = \sigma^2 \mathbb{E}\left[\left\|\left(X^\top L^\top L X/n + \lambda I_p\right)^{-1} n^{-1} X^\top\right\|_F^2\right].$$

Denote $M = \left(X^\top L^\top L X/n + \lambda I_p\right)^{-1}$, the resolvent of the sketched matrix. We further assume that $X$ has iid $\mathcal{N}(0,1)$ entries and $LL^\top = I_m$. Let $L_1$ be an orthogonal complementary matrix of $L$, such that $L^\top L + L_1^\top L_1 = I_n$. We also denote $N = \frac{X^\top L_1^\top L_1 X}{n}$. Then

$$MX^\top X/n = M\frac{X^\top L^\top L X + X^\top L_1^\top L_1 X}{n} = I_p - \lambda M + MN.$$

Therefore, using that $\text{Cov}\left[\beta\right] = \alpha^2/p \cdot I_p$, we find the bias as

$$\text{bias}^2 = \frac{\alpha^2}{p}\mathbb{E}\left[\text{tr}(M - I_p)(M^\top - I_p)\right]$$

$$= \frac{\alpha^2}{p}\left\{\lambda^2\mathbb{E}\left[\text{tr}[M^2]\right] + \mathbb{E}\left[\text{tr}\,M^2\frac{(X^\top L_1^\top L_1 X)^2}{n^2}\right] - 2\lambda\mathbb{E}\left[\text{tr}\,M^2 N\right]\right\}.$$

By the properties of Wishart matrices (e.g., Anderson, 2003; Muirhead, 2009), we have

$$\mathbb{E}\left[N\right] = \frac{n-m}{n}I_p,$$

$$\mathbb{E}\left[(N)^2\right] = \frac{1}{n^2}\mathbb{E}\left[Wishart(I_p, n-m)^2\right] = \frac{1}{n^2}[n - m + p(n-m) + (n-m)^2]I_p.$$

Recalling that $m, n \to \infty$ such that $m/n \to \xi$, and that $\theta_i(\gamma, \lambda) = \int (x + \lambda)^{-i} dF_\gamma(x)$,

$$\text{bias}^2 = \frac{\alpha^2}{p}\left[\lambda^2 + \frac{n - m + p(n-m) + (n-m)^2}{n^2} - 2\lambda\frac{n-m}{n}\right]\mathbb{E}\left[\text{tr}[M^2]\right]$$

$$\to \alpha^2[(\lambda + \xi - 1)^2 + \gamma(1 - \xi)]\theta_2(\gamma, \xi, \lambda).$$

Moreover,

$$\delta^2 = \frac{\sigma^2}{n^2}\mathbb{E}\left[\text{tr}[M^2 X^\top X]\right]$$

$$= \frac{\sigma^2}{n} \cdot \left\{\mathbb{E}\left[\text{tr}[M]\right] - \lambda\mathbb{E}\left[\text{tr}[M^2]\right] + \mathbb{E}\left[\text{tr}[M^2 N]\right]\right\}$$

$$\to \gamma\sigma^2[\theta_1(\gamma, \xi, \lambda) - \lambda\theta_2(\gamma, \xi, \lambda) + (1 - \xi)\theta_2(\gamma, \xi, \lambda)].$$

Here we used the additional definitions

$$\theta_i(\gamma, \xi, \lambda) = \int \frac{1}{(\xi x + \lambda)^i} dF_{\gamma/\xi}(x)$$
$$\theta_i(\gamma, \lambda) = \theta_i(\gamma, \xi = 1, \lambda).$$

Note that these can be connected to the previous definitions by

$$\theta_1(\gamma, \xi, \lambda) = \frac{1}{\xi} \int \frac{1}{x + \lambda/\xi} dF_{\gamma/\xi}(x) = \frac{1}{\xi}\theta_1\left(\frac{\gamma}{\xi}, \frac{\lambda}{\xi}\right)$$
$$\theta_2(\gamma, \xi, \lambda) = \frac{1}{\xi^2}\theta_2\left(\frac{\gamma}{\xi}, \frac{\lambda}{\xi}\right).$$

Therefore the AMSE of $\hat{\beta}_p$ is

$$AMSE(\hat{\beta}_p) = \alpha^2[(\lambda + \xi - 1)^2 + \gamma(1 - \xi)]\theta_2(\gamma, \xi, \lambda) + \gamma\sigma^2[\theta_1(\gamma, \xi, \lambda) - (\lambda + \xi - 1)\theta_2(\gamma, \xi, \lambda)]$$

$$= \alpha^2[(\lambda + \xi - 1)^2 + \gamma(1 - \xi)]\frac{1}{\xi^2}\theta_2\left(\frac{\gamma}{\xi}, \frac{\lambda}{\xi}\right)$$

$$+ \gamma\sigma^2\left[\frac{1}{\xi}\theta_1\left(\frac{\gamma}{\xi}, \frac{\lambda}{\xi}\right) - (\lambda + \xi - 1)\frac{1}{\xi^2}\theta_2\left(\frac{\gamma}{\xi}, \frac{\lambda}{\xi}\right)\right]. \tag{10}$$

$\square$

### A.6.1 ISOTROPIC CASE

Consider the special case where $\Gamma = I$, that is, $X$ has iid $\mathcal{N}(0, 1)$ entries. Then $F_\gamma$ is the standard MP law, and we have the explicit forms for $\theta_i = \theta_i(\gamma, \lambda) = \int \frac{1}{(x+\lambda)^i} dF_\gamma$:

$$\theta_1(\gamma, \lambda) = -\frac{1 + \lambda}{\gamma\lambda} + \frac{1}{2\sqrt{\gamma}\lambda}[\sqrt{\gamma} + \frac{1 + \lambda}{\sqrt{\gamma}} + \sqrt{(\sqrt{\gamma} + \frac{1 + \lambda}{\sqrt{\gamma}})^2 - 4}],$$

$$\theta_2(\gamma, \lambda) = -\frac{1}{\gamma\lambda^2} + \frac{\gamma + 1}{2\gamma\lambda^2} - \frac{1}{2\sqrt{\gamma}}(\frac{\lambda + 1}{\gamma} + 1)\frac{1}{\lambda\sqrt{(\sqrt{\gamma} + \frac{1+\lambda}{\sqrt{\gamma}})^2 - 4}} + \frac{1}{2\sqrt{\gamma}}\sqrt{(\sqrt{\gamma} + \frac{1 + \lambda}{\sqrt{\gamma}})^2 - 4}\frac{1}{\lambda^2},$$

$$\bar{\theta}_1(\zeta, \lambda) = \zeta\theta_1(\zeta, \lambda) + \frac{1 - \zeta}{\lambda},$$

$$\bar{\theta}_2(\zeta, \lambda) = \zeta\theta_2(\zeta, \lambda) + \frac{1 - \zeta}{\lambda^2},$$

The results are obtained by the contour integral formula

$$\int f(x)dF_\gamma(x) = -\frac{1}{4\pi i}\oint_{|z|=1} \frac{f(|1 + \gamma z|^2)(1 - z^2)^2}{z^2(1 + \sqrt{\gamma}z)(z + \sqrt{\gamma})}dz.$$

See Proposition 2.10 of Yao et al. (2015).

### A.7 PROOF OF THEOREM 4.2

*Proof.* Suppose $d/p \to \zeta$ as $n$ goes to infinity. For $\hat{\beta}_d$, we have

$$\text{bias}^2 = \mathbb{E}\left[\left\|n^{-1}X^\top\left(XRR^\top X^\top/n + \lambda I_n\right)^{-1}X\beta - \beta\right\|_2^2\right],$$

$$\delta^2 = \sigma^2 \text{tr}[\left(XRR^\top X^\top/n + \lambda I_n\right)^{-2}\frac{XX^\top}{n^2}].$$

Denote $M = \left(XRR^\top X^\top/n + \lambda I_n\right)^{-1}$. Note that, using that $\text{Cov}[\beta] = \alpha^2/p \cdot I_p$

$$\text{bias}^2 = \frac{\alpha^2}{p}\mathbb{E}\left[\text{tr}[MXX^\top/n]^2\right] - 2\frac{\alpha^2}{p}\mathbb{E}\left[\text{tr}[MXX^\top/n]\right] + \frac{\alpha^2}{p}\text{tr}(I_p).$$

Moreover, letting $R_1$ to be an orthogonal complementary matrix of $R$, such that $RR^\top + R_1 R_1^\top = I_n$, and $N = \frac{XR_1 R_1^\top X^\top}{n}$,

$$\mathbb{E}\left[\frac{1}{p}\operatorname{tr}[MXX^\top/n]\right] = \frac{1}{p}\operatorname{tr}[I_n - \lambda\mathbb{E}\left[\operatorname{tr}[M]\right] + \mathbb{E}\left[MN\right]]$$

$$\rightarrow \frac{1}{\gamma} - \frac{\lambda}{\gamma}\int\frac{1}{x+\lambda}d\bar{F}_\zeta(x) + \frac{\gamma-\zeta}{\gamma}\int\frac{1}{x+\lambda}d\bar{F}_\zeta(x),$$

where $\bar{F}_\zeta$ is the companion MP law, that is, $\bar{F}_\zeta = (1-\gamma)\delta_0 + \gamma F_\zeta$. The third term calculated by using that $XR$ and $XR_1$ are independent for a Gaussian random matrix $X$, so that $M, N$ are independent, and that $\mathbb{E}[N] = \frac{p-d}{n}I_n$. Thus

$$\mathbb{E}\left[\frac{1}{p}\operatorname{tr}[MXX^\top/n]\right] \rightarrow \frac{1}{\gamma} - \frac{\lambda+\zeta-\gamma}{\gamma}\bar{\theta}_1(\zeta,\lambda)$$

$$= \frac{1}{\gamma} - \frac{\lambda+\zeta-\gamma}{\gamma}\left[\frac{1-\zeta}{\lambda} + \zeta\theta_1(\zeta,\lambda)\right].$$

Then

$$\mathbb{E}\left[\frac{1}{p}\operatorname{tr}[MXX^\top/n]^2\right] = \frac{1}{p}\mathbb{E}\left[\operatorname{tr}[I_n + \lambda^2 M^2 + MNMN - 2\lambda M + 2MN - \lambda M^2 N - \lambda MNM]\right].$$

Note that

$$\mathbb{E}[MNMN|M] = M[(p-d)(M^\top + \operatorname{tr}(M)I_n) + (p-d)^2 M]/n^2$$

$$= \frac{p-d+(p-d)^2}{n^2}M^2 + \frac{p-d}{n^2}\operatorname{tr}(M)M,$$

so

$$\mathbb{E}\left[\frac{1}{p}\operatorname{tr}[MXX^\top/n]^2\right] \rightarrow \frac{1}{\gamma}[1 + (\lambda^2 - 2\lambda(\gamma-\zeta) + (\gamma-\zeta)^2)\bar{\theta}_2(\zeta,\lambda)$$

$$+ 2(\gamma-\zeta-\lambda)\bar{\theta}_1(\zeta,\lambda) + (\gamma-\zeta)\bar{\theta}_1^2(\zeta,\lambda)].$$

Thus we find the following exprssion for the limiting squared bias:

$$\text{bias}^2 \rightarrow \frac{\alpha^2}{\gamma}[\gamma - 1 + (\lambda - \gamma + \zeta)^2\bar{\theta}_2 + (\gamma-\zeta)\bar{\theta}_1^2].$$

With similar calculations (that we omit for brevity), we can find

$$\delta^2 \rightarrow \sigma^2(\bar{\theta}_1(\zeta,\lambda) - (\lambda+\zeta-\gamma)\bar{\theta}_2(\zeta,\lambda)).$$

Therefore the AMSE of $\hat{\beta}_d$ is

$$AMSE = \frac{\alpha^2}{\gamma}[\gamma - 1 + (\lambda - \gamma + \zeta)^2\bar{\theta}_2 + (\gamma-\zeta)\bar{\theta}_1^2] + \sigma^2[\bar{\theta}_1(\zeta,\lambda) - (\lambda+\zeta-\gamma)\bar{\theta}_2(\zeta,\lambda)].$$

(11)

$\square$

### A.8   PROOF OF THEOREM 4.3

*Proof.* Recall that we have $m, n \rightarrow \infty$, such that $m/n \rightarrow \xi$. Then we need to take $\xi \rightarrow 0$. However, we find it more convenient to do the calculation directly from the finite sample results as $m, n, p \rightarrow \infty$ with $m/n \rightarrow 0$, $p/n \rightarrow \gamma$, It is not hard to check that computing the results in the other way (i.e., interchanging the limits), leads to the same results. Starting from our bias formula for primal sketching, we first get

$$\text{bias}^2 = \frac{\alpha^2}{p}\left[\lambda^2 + \frac{n-m+p(n-m)+(n-m)^2}{n^2} - 2\lambda\frac{n-m}{n}\right]\mathbb{E}\left[\operatorname{tr}[(X^\top L^\top L X/n + \lambda I_p)^{-2}]\right]$$

$$\rightarrow \alpha^2[(\lambda-1)^2 + \gamma]/\lambda^2.$$

The limit of the trace term is not entirely trivial, but it can be calculated by (1) observing that the $m \times p$ sketched data matrix $P = LX$ has iid normal entries (2) thus the operator norm of $P^\top P/n$ vanishes, (3) and so by a simple matrix perturbation argument the trace concentrates around $p/\lambda^2$. This gives the rough steps of finding the above limit. Moreover,

$$\delta^2 = \frac{\sigma^2}{n^2}\mathbb{E}\left[\text{tr}[(X^\top L^\top LX/n + \lambda I_p)^{-2} X^\top X]\right] \to \gamma\sigma^2/\lambda^2 \cdot \mathbb{E}_{F_\gamma} X^2 = \gamma\sigma^2/\lambda^2$$

So the MSE is $M(\lambda) = \alpha^2[(\lambda - 1)^2 + \gamma]/\lambda^2 + \sigma^2 \cdot \gamma/\lambda^2$. From this it is elementary to find the optimal $\lambda$ and its objective value. $\qquad\square$

### A.9 PROOF OF THEOREM 4.4

*Proof.* Note that the bias can be written as

$$\text{bias}^2 = \frac{\alpha^2}{p}\mathbb{E}\left[\text{tr}\left[\left(\frac{XRR^\top X^\top}{nd} + \lambda I_n\right)^{-1}\frac{XX^\top}{nd}\right]^2\right]$$

$$- 2\frac{\alpha^2}{p}\mathbb{E}\left[\text{tr}[(XRR^\top X^\top/n + \lambda I_n)^{-1} XX^\top/n]\right] + \alpha^2.$$

Write $G = XX^\top$. Since $RR^\top \sim \mathcal{W}_p(I_p, d)$, we have $XRR^\top X^\top \sim \mathcal{W}_n(G, d)$. So $XRR^\top X^\top \stackrel{d}{=} G^{1/2}WG^{1/2}$, where $W \sim \mathcal{W}_n(I_n, d)$.

$$\mathbb{E}\left[\text{tr}\left[\left(\frac{XRR^\top X^\top}{nd} + \lambda I_n\right)^{-1} XX^\top/n\right]\right] = \mathbb{E}\left[\text{tr}[(G^{1/2}WG^{1/2}/d + n\lambda I_n)^{-1}G]\right]$$

$$= \mathbb{E}\left[\text{tr}[(\frac{W}{d} + \lambda(\frac{G}{n})^{-1})^{-1}]\right].$$

So we need to find the law of $\frac{W}{d} + \frac{\lambda}{\gamma}(\frac{G}{p})^{-1}$. Suppose first that $G = XX^\top \sim \mathcal{W}_n(I_n, p)$. Then $W$ and $G^{-1}$ are asymptotically freely independent. The l.s.d. of $W/d$ is the MP law $F_{1/\xi}$ while the l.s.d. of $G/p$ is the MP law $F_{1/\gamma}$. We need to find the additive free convolution $W \boxplus \bar{G}$, where $\bar{G} = \frac{\lambda}{\gamma}G^{-1}$.

Recall that the $R$-transform of a distribution $F$ is defined by

$$R_F(z) = m_F^{-1}(-z) - \frac{1}{z},$$

where $m_F^{-1}(z)$ is the inverse function of the Stieltjes transform of $F$ (e.g., Voiculescu et al., 1992; Hiai & Petz, 2006; Couillet & Debbah, 2011). We can find the $R$-transform by solving

$$m_F(R_F(z) + \frac{1}{z}) = -z.$$

Note that the $R$-transform of $W/d$ is

$$R_W(z) = \frac{1}{1 - z/\xi}.$$

The Stieltjes transform of $G^{-1}$ is

$$m_{G^{-1}}(z) = \int \frac{1}{1/x - z}dF_{1/\gamma}(x) = -\frac{1}{z} - \frac{1}{z^2}m_{1/\gamma}(\frac{1}{z})$$

$$= -\frac{1}{z} - \frac{1 - \frac{1}{\gamma} - \frac{1}{z} + \sqrt{(1 + \frac{1}{\gamma} + \frac{1}{z})^2 - \frac{4}{\gamma}}}{2\frac{z}{\gamma}}$$

$$= -\frac{1 + \frac{1}{\gamma} - \frac{1}{z} + \sqrt{(1 + \frac{1}{\gamma} - \frac{1}{z})^2 - \frac{4}{\gamma}}}{2\frac{z}{\gamma}}.$$

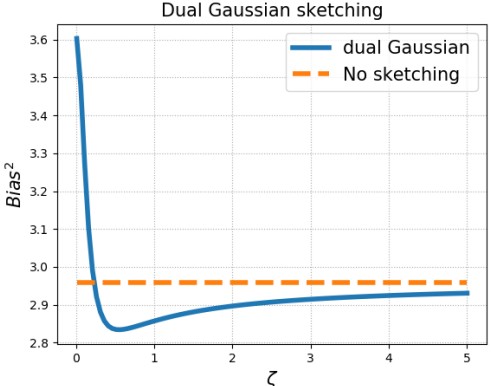

Figure 9: Dual Gaussian sketch improves MSE.

Then the $R$-transform of $G^{-1}$ is

$$R_{G^{-1}}(z) = -\frac{1}{z} + \frac{\gamma + 1 - \sqrt{(\gamma+1)^2 - 4\gamma(z+1)}}{2z}$$

$$= \frac{\gamma - 1 - \sqrt{(\gamma-1)^2 - 4\gamma z}}{2z}.$$

Since we have the property that $R_{a\mu}(z) = aR_\mu(az)$,

$$R_{\bar{G}} = R_{\frac{\lambda}{\gamma}G^{-1}}(z) = \frac{\gamma - 1 - \sqrt{(\gamma-1)^2 - 4\lambda z}}{2z}.$$

Hence we have

$$R_{W \boxplus \bar{G}} = R_W + R_{\bar{G}} = \frac{1}{1 - z/\xi} + \frac{\gamma - 1 - \sqrt{(\gamma-1)^2 - 4\lambda z}}{2z}.$$

Moreover, the Stieltjes transform of $\mu = W \boxplus \bar{G}$ satisfies

$$m_\mu^{-1}(z) = m_{W \boxplus \bar{G}}^{-1}(z) = R_F(-z) - \frac{1}{z} = \frac{1}{1 + z/\xi} + \frac{\gamma - 1 - \sqrt{(\gamma-1)^2 + 4\lambda z}}{-2z} - \frac{1}{z}.$$

Note that

$$2\frac{\alpha^2}{p} \mathbb{E}\left[ \text{tr}\left[ \left( \frac{XRR^\top X^\top}{nd} + \lambda I_n \right)^{-1} XX^\top/n \right] \right] \to 2\frac{\alpha^2}{\gamma} \mathbb{E}_\mu\left[\frac{1}{x}\right] = 2\frac{\alpha^2}{\gamma} \lim_{z \to 0} m(z),$$

$$\frac{\alpha^2}{p} \mathbb{E}\left[ \text{tr}\left[ \left( \frac{XRR^\top X^\top}{nd} + \lambda I_n \right)^{-1} \frac{XX^\top}{nd} \right]^2 \right] \to \frac{\alpha^2}{\gamma} \mathbb{E}_\mu\left[\frac{1}{x^2}\right] = \frac{\alpha^2}{\gamma} \lim_{z \to 0} \frac{d}{dz} m(z).$$

So it suffices to find $m(z)$ and $\frac{d}{dz}m(z)$ evaluated at zero. $\qquad\square$

This result can characterize the performance of sketching in the high SNR regime, where $\alpha \gg \sigma$. To understand the lower SNR regime, we need to study the variance, and thus we need to calculate

$$\text{var} = \sigma^2 \frac{1}{n} \mathbb{E}\left[ \text{tr}\left[ \left( \frac{XRR^\top X^\top}{nd} + \lambda I_n \right)^{-2} XX^\top/n \right] \right] = \sigma^2 \mathbb{E}\left[ \text{tr}[(\frac{W}{d} + \frac{\lambda}{\gamma}(\frac{G}{p})^{-1})^{-2}G^{-1}] \right]$$

where $G = XX^\top \sim \mathcal{W}_n(I_n, p)$ is a Wishart distribution, and $XRR^\top X^\top =_d G^{1/2}WG^{1/2}$, with $W \sim \mathcal{W}_n(I_n, r)$. This seems to be quite challenging, and we leave it to future work.

A.10 RESULTS FOR PRIMAL GAUSSIAN SKETCHING

The statement requires some notions from free probability, see e.g., Voiculescu et al. (1992); Hiai & Petz (2006); Nica & Speicher (2006); Anderson et al. (2010); Couillet & Debbah (2011) for references .

**Theorem A.1** (Bias of primal Gaussian sketch). *Suppose $X$ is an $n \times p$ standard Gaussian random matrix. Suppose also that $L$ is a $d \times n$ matrix with i.i.d. $\mathcal{N}(0, 1/d)$ entries. Then the bias of primal sketch has the expression $MSE(\hat{\beta}_p) = \alpha^2 + \frac{\alpha^2}{\gamma}[\tau((a+b)^{-1}b(a+b)^{-1}b^{-1}) - 2\tau((a+b)^{-1})]$, where $a$ and $b$ two free random variables, that are freely independent in a non-commutative probability space, and $\tau$ is their trace. Specifically, the law of $a$ is the MP law $F_{1/\xi}$ and $b = \frac{\lambda}{\gamma}\tilde{b}$, where the law of $\tilde{b}$ is the MP law $F_{1/\gamma}$.*

*Proof of Theorem A.1.* Note that

$$\text{bias}^2 = \mathbb{E}\left[\left\|\left(X^\top L^\top L X/(nd) + \lambda I_p\right)^{-1}(X^\top X/n)\beta - \beta\right\|_2^2\right],$$

and $\left(X^\top L^\top L X/(nd) + \lambda I_p\right)^{-1} X^\top = X^\top (L^\top L X X^\top/(nd) + \lambda I_n)^{-1}$. Thus

$$\text{bias}^2 = \mathbb{E}\left[\left\|X^\top(\frac{L^\top L X X^\top}{nd} + \lambda I_n)^{-1}\frac{X}{n}\beta - \beta\right\|_2^2\right]$$

$$= \alpha^2 + \frac{\alpha^2}{p}\mathbb{E}[\text{tr}[(\frac{XX^\top L^\top L}{nd} + \lambda I_n)^{-1}\frac{XX^\top}{n}(\frac{L^\top L X X^\top}{nd} + \lambda I_n)^{-1}\frac{XX^\top}{n}]$$

$$- 2\,\text{tr}[(\frac{L^\top L X X^\top}{nd} + \lambda I_n)^{-1}\frac{XX^\top}{n}]].$$

First we find the l.s.d. of $(\frac{L^\top L X X^\top}{nd} + \lambda I_n)^{-1}\frac{XX^\top}{n}$. Write $W = L^\top L$, $G = XX^\top$. Then

$$(\frac{L^\top L X X^\top}{nd} + \lambda I_n)^{-1}\frac{XX^\top}{n} = (\frac{WG}{nd} + \lambda I_n)^{-1}\frac{G}{n} = G^{-1}(\frac{W}{d} + \lambda(\frac{G}{n})^{-1})^{-1}G,$$

which is similar to $(\frac{W}{d} + \lambda(\frac{G}{n})^{-1})^{-1}$. So it suffices to find the l.s.d. of $(\frac{W}{d} + \frac{\lambda}{\gamma}(\frac{G}{p})^{-1})^{-1}$.

By the definition, $W \sim \mathcal{W}_n(I_n, d)$, $G \sim \mathcal{W}_n(I_n, p)$, therefore the l.s.d. of $W/d$ converges to the MP law $F_{1/\xi}$ and the l.s.d. of $G/p$ converges to the MP law $F_{1/\gamma}$.

Also note that

$$(\frac{XX^\top L^\top L}{nd} + \lambda I_n)^{-1}\frac{XX^\top}{n}(\frac{L^\top L X X^\top}{nd} + \lambda I_n)^{-1}\frac{XX^\top}{n} = (\frac{W}{d} + \lambda n G^{-1})^{-1}G^{-1}(\frac{W}{d} + \lambda n G^{-1})^{-1}G.$$

We write $A = \frac{W}{d}$, $B = \frac{\lambda}{\gamma}(\frac{G}{p})^{-1}$. Then it suffices to find

$$\frac{\alpha^2}{p}\mathbb{E}\left[\text{tr}[(A + B)^{-1}B(A + B)^{-1}B^{-1}]\right].$$

We will find an expression for this using free probability. For this we will need to use some series expansions. There are two cases, depending on whether the operator norm of $BA^{-1}$ is less than or greater than unity, leading to different series expansions. We will work out below the first case, but the second case is similar and leads to the same answer.

$$\text{tr}[(A + B)^{-1}B(A + B)^{-1}B^{-1}] = \text{tr}[A^{-1}(I + BA^{-1})^{-1}BA^{-1}(I + BA^{-1})^{-1}B^{-1}]$$

Since the operator norm of $BA^{-1}$ is less unity, we have the von Neumann series expansion

$$[I + BA^{-1}]^{-1} = \sum_{i=0}^{\infty}(-BA^{-1})^i,$$

then we have

$$\text{tr}[(A+B)^{-1}B(A+B)^{-1}B^{-1}] = \sum_{i,j\geq 0}(-1)^{i+j}\text{tr}[(BA^{-1})^{i+j+1}B^{-1}A^{-1}]$$

$$= \sum_{i,j\geq 0}(-1)^{i+j}\text{tr}[(A^{-1}B)^{i+j+1}A^{-1}B^{-1}].$$

Since $A$ and $B$ are asymptotically freely independent in the free probability space arising in the limit (e.g., Voiculescu et al., 1992; Hiai & Petz, 2006; Couillet & Debbah, 2011), and the polynomial $(a^{-1}b)^{i+j+1}a^{-1}b^{-1}$ involves an alternating sequence of $a, b$, we have

$$\frac{1}{n}\text{tr}[(A^{-1}B)^{i+j+1}A^{-1}B^{-1}] \to \tau[(a^{-1}b)^{i+j+1}a^{-1}b^{-1}],$$

where $a$ and the $b$ are free random variables and $\tau$ is their law. Specifically, $a$ is a free random variable with the MP law $F_{1/\xi}$ and $b$ is $\frac{\lambda}{\gamma}\tilde{b}^{-1}$, where $\tilde{b}$ is a free r.v. with MP law $F_{1/\gamma}$. Moreover, they are freely independent.

Hence, we have

$$\frac{1}{n}\text{tr}[(A+B)^{-1}B(A+B)^{-1}B^{-1}] \to \tau\Big[\sum_{i\geq 0}(-1)^i(a^{-1}b)^{i+1}\sum_{j\geq 0}(-1)^j(a^{-1}b)^ja^{-1}b^{-1}\Big]$$

$$= \tau[(a^{-1}b)(1+a^{-1}b)^{-1}(1+a^{-1}b)^{-1}a^{-1}b^{-1}]$$

$$= \tau[(a+b)^{-1}b(a+b)^{-1}b^{-1}].$$

Therefore,

$$\text{bias}^2 \to \alpha^2 + \frac{\alpha^2}{\gamma}\left[\frac{1}{n}\text{tr}[(A+B)^{-1}B(A+B)^{-1}B^{-1} - 2\text{tr}[A+B]^{-1}]\right]$$

$$= \alpha^2 + \frac{\alpha^2}{\gamma}[\tau((a+b)^{-1}b(a+b)^{-1}b^{-1}) - 2\tau((a+b)^{-1})].$$

$\square$

## A.11 RESULTS FOR FULL SKETCHING

The full sketch estimator projects down the entire data, and then does ridge regression on the sketched data. It has the form

$$\hat{\beta}_f = \left(X^\top L^\top LX/n + \lambda I_p\right)^{-1}\frac{X^\top L^\top LY}{n}.$$

We have

$$\text{bias}^2 = \alpha^2\lambda^2\int\frac{1}{\xi x+\lambda}dF_{\gamma/\xi}(x) = \alpha^2\lambda^2\frac{1}{\xi^2}\theta_2\left(\frac{\gamma}{\xi},\frac{\lambda}{\xi}\right)$$

$$\text{var} = \sigma^2\gamma\left[\int\frac{1}{\xi x+\lambda}dF_{\gamma/\xi}(x) - \lambda\int\frac{1}{(\xi x+\lambda)^2}dF_{\gamma/\xi}(x)\right]$$

$$= \sigma^2\gamma\left[\frac{1}{\xi}\theta_1\left(\frac{\gamma}{\xi},\frac{\lambda}{\xi}\right) - \lambda\frac{1}{\xi^2}\theta_2\left(\frac{\gamma}{\xi},\frac{\lambda}{\xi}\right)\right],$$

therefore

$$AMSE(\hat{\beta}_f) = \alpha^2\lambda^2\frac{1}{\xi^2}\theta_2\left(\frac{\gamma}{\xi},\frac{\lambda}{\xi}\right) + \sigma^2\gamma\left[\frac{1}{\xi}\theta_1\left(\frac{\gamma}{\xi},\frac{\lambda}{\xi}\right) - \lambda\frac{1}{\xi^2}\theta_2\left(\frac{\gamma}{\xi},\frac{\lambda}{\xi}\right)\right]$$

The optimal $\lambda$ for full sketch is always $\lambda^* = \frac{\gamma\sigma^2}{\alpha^2}$, the same as ridge regression. Some simulation results are shown in Figure 10, and they show the expected shape (e.g., they decrease with $\xi$).

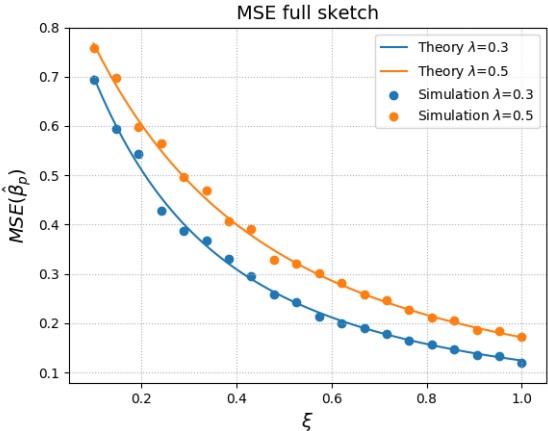

Figure 10: Simulation results for full sketch, with $n = 1000, \gamma = 0.1$. The simulation results are averaged over 30 independent experiments.

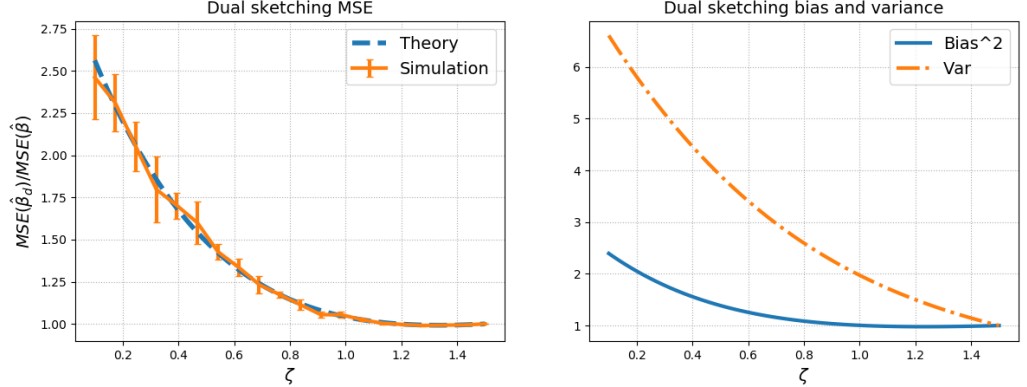

Figure 11: Dual orthogonal sketching with $\gamma = 1.5, \lambda = 1, \alpha = 3, \sigma = 1$. Left: MSE of dual sketching normalized by the MSE of ridge regression. The standard deviation is over 50 repetitions. Right: Bias and variance of dual sketching normalized by the bias and variance of ridge regression, respectively.

## A.12 NUMERICAL RESULTS

### A.12.1 DUAL ORTHOGONAL SKETCHING

See Figure 11 for additional simulation results for dual orthogonal sketching.

### A.12.2 PERFORMANCE AT A FIXED REGULARIZATION PARAMETER

First we fix the regularization parameter at the optimal value for original ridge regression. The results are visualized in Figure 12. On the $x$ axis, we plot the reduction in sample size $m/n$ for primal sketch, and the reduction in dimension $d/p$ for dual sketch. In this case, primal and dual sketch will increase both bias and variance, and empirically in the current case, dual sketch increases them more. So in this particular case, primal sketch is preferred.

### A.12.3 PERFORMANCE AT THE OPTIMAL REGULARIZATION PARAMETER

We find the optimal regularization parameter $\lambda$ for primal and dual orthogonal sketching. Then we use the optimal regularization parameter for all settings, see Figure 13. Both primal and dual sketch increase the bias, but decrease the variance. It is interesting to note that, for equal parameters $\xi$ and

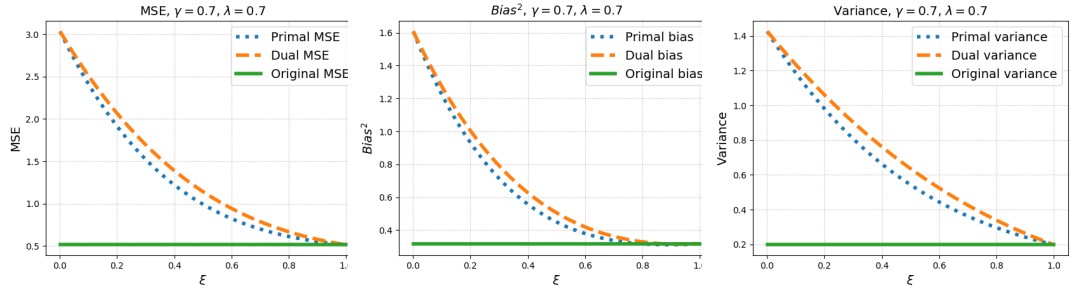

Figure 12: Fixed regularization parameter $\lambda = 0.7$, optimal for original ridge, in a setting where $\gamma = 0.7$, and $\alpha^2 = \sigma^2$.

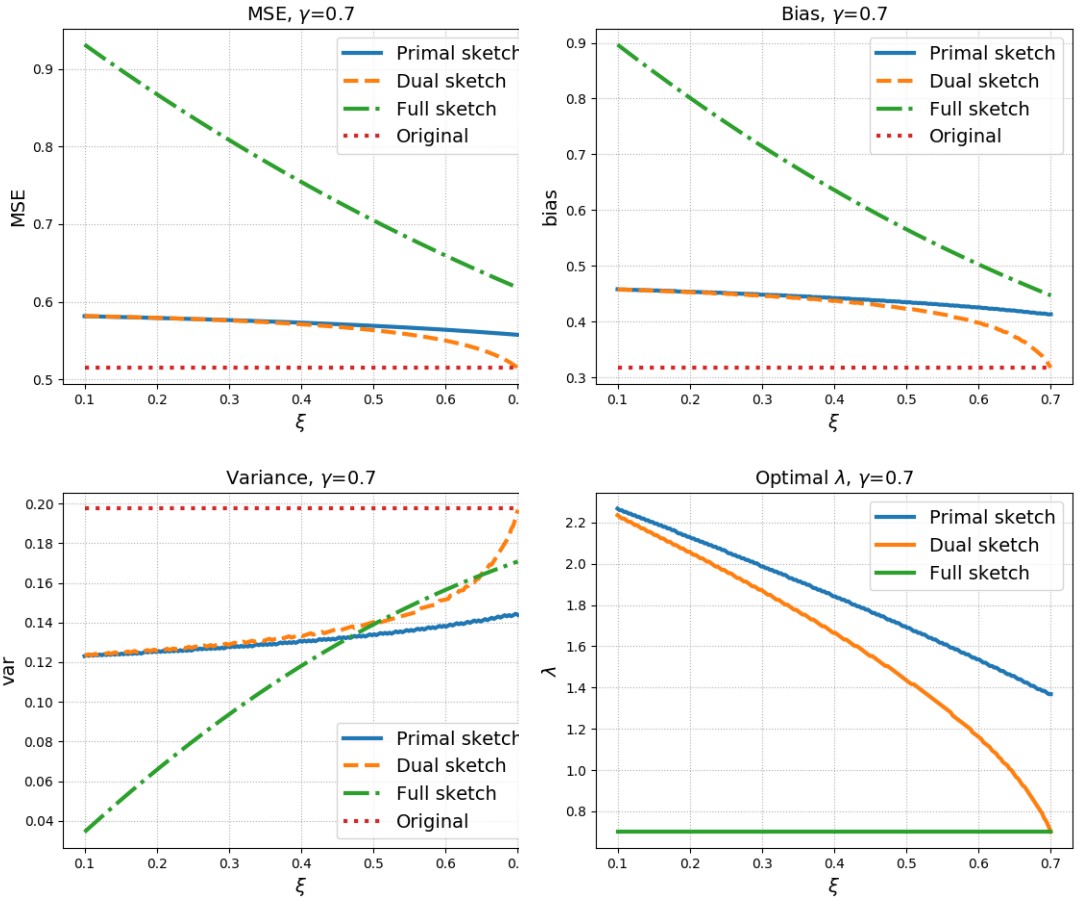

Figure 13: Primal and dual sketch at optimal $\lambda$. We take $\gamma = 0.7$ and let $\xi$ range between 0.001 and 1, where for primal sketch $\xi = r/n$ while for dual sketch $\xi = d/p$.

$\zeta$, and in our particular case, dual sketch has smaller variance, but larger bias. So primal sketch is preferred bias or MSE is important, but dual sketch is more desired when one wants smaller variance. All in all, dual sketch has larger MSE than primal sketch in the current setting. It can also be seen that in this specific example, the optimal $\lambda$ for primal sketch is smaller than that of dual sketch. However these results are hard to interpret, because there is no natural correspondence between the two parameters $\xi$ and $\zeta$.

## A.13    COMPUTATIONAL COMPLEXITY

Since sketching is a method to reduce computational complexity, it is important to discuss how much computational efficiency we gain. Recall our three estimators

$$
\hat{\beta} = \left(X^\top X/n + \lambda I_p\right)^{-1} X^\top Y/n = n^{-1} X^\top \left(XX^\top/n + \lambda I_n\right)^{-1} Y,
$$
$$
\hat{\beta}_p = \left(X^\top L^\top L X/n + \lambda I_p\right)^{-1} X^\top Y/n,
$$
$$
\hat{\beta}_d = n^{-1} X^\top \left(XRR^\top X^\top/n + \lambda I_n\right)^{-1} Y,
$$

Their computational complexity, when computed in the usual way, is:

- No sketch (Standard ridge): if $p < n$, computing $X^\top Y$ and $X^\top X$ requires $O(np)$ and $O(np^2)$ flops, then solving the linear equation $(X^\top X/n + \lambda I_p)\hat{\beta} = X^\top Y/n$ requires $O(p^3)$ flops by the LU decomposition. It is $O(np^2)$ flops in total.

  If $p > n$, we use the second formula for $\hat{\beta}$, and the total flops is $O(pn^2)$.

- Primal sketch: for the Hadamard sketch (and other sketches based on the FFT), computing $LX$ by FFT requires $mp \log n$, computing $(LX)^\top LX$ requires $mp^2$, so the total flops is $O(p^3 + mp(\log n + p))$. So the primal sketch can reduce the computation cost only when $p < n$.

- Dual sketch: computing $XRR^\top X^\top$ requires $nd\,(\log p + n)$ flops by FFT, solving $(XRR^\top X^\top/n + \lambda I_n)^{-1} Y$ requires $O(n^3)$ flops, the matrix-vector multiplication of $X^\top$ and $(XRR^\top X^\top/n + \lambda I_n)^{-1} Y$ requires $O(np)$ flops, so the total flops is $O(n^3 + nd(\log p + n))$. Dual sketching can reduce the computation cost only when $p > n$.

