# OpenReview forum: "Ridge Regression: Structure, Cross-Validation, and Sketching"
_ICLR.cc/2020/Conference — Accept (Spotlight)_

### Official Review · AnonReviewer2 · 2019-10-20
**Official Blind Review #2**

**Rating:** 8

**Review:**

This paper presents a theoretical study of ridge regression, focusing on the practical problems of correcting for the bias of the cross-validation based estimate of the optimal regularisation parameter, and quantification of the asymptotic risk of sketching algorithms for ridge regression, both in the p / n -> gamma in (0, 1) regime (n = # data points, p = # dimensions). The authors derive most of their results exploiting their (AFAICT) new asymptotic characterisation of the ridge regression estimator which may be of independent interest. The whole study is complemented by a series of numerical experiments.

I am recommending this paper to be accepted for publication at ICLR. The paper is clearly written, makes several solid theoretical contributions, and recommends a simple and practical bias correction for CV-based estimates of the optimal ridge regulariser. While ICLR is biased towards deep learning focused publications, the work of Belkin, Hsu, Ma, Bartlett, Hastie, Montanari, Rakhlin, Liang and others (apologies to everyone whose name was omitted---it is for the sole reason of brevity) has recently shown that we can learn non-negligible amount about neural networks from study of linear models.


Comments:

- Most of the examples in the paper focus on the regime gamma < 1. Would you expect the observed behaviours to be significantly different when gamma > 1? I am asking specifically because of [1] which has found that the bias of the risk estimate obtained via cross-validation is often most extreme when p >> n, which makes me wonder about what would an experiment like those in fig.2 look like in the p >> n regime?

- I was somewhat confused when I first read the statement of thm.2.1. In particular, the definition of asymptotic equivalence requires (roughly speaking) that any series of projections of the difference between the two random vectors converges to zero a.s. However, within the theorem, you introduce Z without much explanation which confused me because not every Z ~ standard normal would be asymptotically equivalent. I needed to look at the proof to understand how Z is “coupled” with hat(beta), which I think should not be necessary. If possible, I would either say that there exists a (series of) Z (all standard normal) s.t. the asymptotic equivalence holds, or add some other clarification (possibly in the form of a footnote).

- When you are citing a book, please consider citing exact pages or at least chapters/sections (e.g., when citing the exact shortcut from Hastie et al. (2019)).

- In the definition of asymptotic equivalence (starting at the bottom of p.2), did you mean to assume limsup ||w|| < \infty a.s. (or is limsup not needed here)?


References:

[1] Tibshirani, R. J., & Tibshirani, R. (2009). A bias correction for the minimum error rate in cross-validation. The Annals of Applied Statistics, 822-829.

**Experience Assessment:**

I do not know much about this area.

**Review Assessment: Checking Correctness Of Derivations And Theory:**

I assessed the sensibility of the derivations and theory.

**Review Assessment: Checking Correctness Of Experiments:**

I assessed the sensibility of the experiments.

**Review Assessment: Thoroughness In Paper Reading:**

I read the paper at least twice and used my best judgement in assessing the paper.

---

> ### Author Response · Authors · 2019-11-10
> **Reply to Referee #2**
>
> We thank the referee for the many helpful comments. We provide some answers below.
>
>
> "This paper presents a theoretical study of ridge regression, focusing on the practical problems of correcting for the bias of the cross-validation based estimate of the optimal regularisation parameter, and quantification of the asymptotic risk of sketching algorithms for ridge regression, both in the p / n -> gamma in (0, 1) regime (n = # data points, p = # dimensions). The authors derive most of their results exploiting their (AFAICT) new asymptotic characterisation of the ridge regression estimator which may be of independent interest. The whole study is complemented by a series of numerical experiments."
>
> - Thanks for that summary. We would like to clarify that the asymptotic characterization (while new), can only be used to derive part of our results. Namely, the representation characterizes linear functionals of the estimator, and thus can be use to derive the bias of the estimator. However, it can not directly be used to derive the variance, and in fact this seems to require some major advances in random matrix theory. We have added a comment about this after the theorem.
>
> Moreover, we would like to clarify that we allow gamma>1 everywhere in the paper. This is possible because we work with a positive lambda>0 regularization parameter, and so ridge regression is always well defined.
>
>
> "I am recommending this paper to be accepted for publication at ICLR. The paper is clearly written, makes several solid theoretical contributions, and recommends a simple and practical bias correction for CV-based estimates of the optimal ridge regulariser. While ICLR is biased towards deep learning focused publications, the work of Belkin, Hsu, Ma, Bartlett, Hastie, Montanari, Rakhlin, Liang and others (apologies to everyone whose name was omitted---it is for the sole reason of brevity) has recently shown that we can learn non-negligible amount about neural networks from study of linear models."
>
>
> Comments:
>
> "- Most of the examples in the paper focus on the regime gamma < 1. Would you expect the observed behaviours to be significantly different when gamma > 1? I am asking specifically because of [1] which has found that the bias of the risk estimate obtained via cross-validation is often most extreme when p >> n, which makes me wonder about what would an experiment like those in fig.2 look like in the p >> n regime?"
>
> - As we mentioned above, our theory covers gamma>1. For Figure 2, the particular datasets we used happen to have n>p (ie gamma<1). We have added some additional simulations for the p>>n case to the already existing simulations, see Figure 7 in Section A.5. However, in this particular case, while CV indeed has a very large bias for the test error, our correction does not significantly improve the test error. Looking at the figure, it seems we would need a stronger bias correction; however, it is unclear to us at the moment how to od this in a principled way.
>
> "- I was somewhat confused when I first read the statement of thm.2.1. In particular, the definition of asymptotic equivalence requires (roughly speaking) that any series of projections of the difference between the two random vectors converges to zero a.s. However, within the theorem, you introduce Z without much explanation which confused me because not every Z ~ standard normal would be asymptotically equivalent. I needed to look at the proof to understand how Z is “coupled” with hat(beta), which I think should not be necessary. If possible, I would either say that there exists a (series of) Z (all standard normal) s.t. the asymptotic equivalence holds, or add some other clarification (possibly in the form of a footnote)."
>
> - We have clarified the statement and mentioned (right after the theorem) that the noise is coupled with the estimator. However, we don't think that the result would hold for an independent noise vector Z, because that would require the two noise vectors Z,Z' to be asymptotically equivalent, while we think that they cannot be (e.g., the difference between the first coordinates is O(1) and does not vanish.)
>
>
> "- When you are citing a book, please consider citing exact pages or at least chapters/sections (e.g., when citing the exact shortcut from Hastie et al. (2019))."
>
> - Actually the cited reference Hastie et al 2019 is a paper ("surprises in interpolation"). We already had the reference to page 243 of Hastie et al 2009 (the ESL book).
>
> "- In the definition of asymptotic equivalence (starting at the bottom of p.2), did you mean to assume limsup ||w|| < \infty a.s. (or is limsup not needed here)?"
>
> Good point, we added the limsup.

---

### Official Review · AnonReviewer1 · 2019-10-22
**Official Blind Review #1**

**Rating:** 6

**Review:**

This paper deals with 3 theoretical properties of ridge regression. First, it proves that the ridge regression estimator is equivalent to a specific representation which is useful as for instance it can be used to derive the training error of the ridge estimator. Second, it provides a bias correction mechanism for ridge regression and finally it provides proofs regarding the accuracy of several sketching algorithms for ridge regression.

The paper addresses an important problem and puts itself nicely in context of previous work. However, it comes across as three papers stapled together, that were submitted to some journal and have now been put into ICLR format. Most of the important results are in the appendix. The main body of the paper is just a smattering of theorems with some text flowing around them and too much notation. There is little or no intuition provided for the proofs in the paper. Moreover, the connection between the 3 theoretical properties of ridge regression studied is also unclear. There could very well be 2 or 3 conference papers written out of this one paper.

As far as the technical merit is concerned, I checked some theory and it appears correct, however some things were unclear. For example, the Theorem 2.1 is proven under a random design setting; how would the proven ridge representation look under a fixed design setting? Or can we even prove something in that case?

I think the paper definitely has some merit but the presentation makes it hard to assess it.


**Experience Assessment:**

I have published one or two papers in this area.

**Review Assessment: Checking Correctness Of Derivations And Theory:**

I assessed the sensibility of the derivations and theory.

**Review Assessment: Checking Correctness Of Experiments:**

I carefully checked the experiments.

**Review Assessment: Thoroughness In Paper Reading:**

I read the paper at least twice and used my best judgement in assessing the paper.

---

> ### Author Response · Authors · 2019-11-10
> **Reply to Reviewer #1**
>
> We thank the referee for their careful reading of the manuscript and for the many helpful suggestions. We have made several changes to try and address the issues they raised. We reproduce their comments and our replies below. We are more than happy to work with the referee to make sure we address all comments within our ability.
>
>
> "This paper deals with 3 theoretical properties of ridge regression. First, it proves that the ridge regression estimator is equivalent to a specific representation which is useful as for instance it can be used to derive the training error of the ridge estimator. Second, it provides a bias correction mechanism for ridge regression and finally it provides proofs regarding the accuracy of several sketching algorithms for ridge regression."
>
> - We thank the reviewer for this summary.
>
> "The paper addresses an important problem and puts itself nicely in context of previous work. However, it comes across as three papers stapled together, that were submitted to some journal and have now been put into ICLR format."
>
> - We thank the reviewer for acknowledging the importance of the problem we address. However, we emphasize that we have *not* submitted this paper anywhere (neither to a journal nor to a conference) before, and so this is perhaps why the structure is less than perfect. However, we believe that the unifying theme of our paper is that we study ridge regression in a common high-dimensional asymptotic framework, and also several results rely on having a random effects model for the regression coefficients beta. We think that this gives the paper a unifying theme.
>
> "Most of the important results are in the appendix."
>
> - We thank the reviewer for raising this concern. We have moved one of the results from the appendix to the main body, namely the result on the MSE and training error of ridge (now Theorem 2.2). We think that most of the other results in the appendix are now merely supporting results, and the main results are already in the paper. However, if the referee thinks there are some remaining results which should be moved to the main paper, we are happy to do so (please let us know which ones.)
>
> "The main body of the paper is just a smattering of theorems with some text flowing around them and too much notation. There is little or no intuition provided for the proofs in the paper."
>
> - We have attempted to add more text in the main body, and in particular increased the number of pages from 8 to 10. We have added paragraph headings to many places, and we hope that this will make the paper easier to navigate. We have also added intuitive descriptions (and high-level steps) to the proofs, right after the statements. We hope that this will make the paper more readable, and please let us know what else we should improve.
>
> "Moreover, the connection between the 3 theoretical properties of ridge regression studied is also unclear. There could very well be 2 or 3 conference papers written out of this one paper. "
>
> - We agree that there are a few separate topics addressed here. However, as we mentioned above, we think that they are tied together by the common high-dimensional asymptotic framework. In terms of structuring, we hope that this structure will make one strong paper, instead of two-three not so strong ones. However, if the referee insists, we can try to restructure our paper (e.g., move sketching to a separate paper).
>
> "As far as the technical merit is concerned, I checked some theory and it appears correct, however some things were unclear. For example, the Theorem 2.1 is proven under a random design setting; how would the proven ridge representation look under a fixed design setting? Or can we even prove something in that case?"
>
> - Thank you for that insightful question. The representation indeed depends on the random design setting. For a fixed design, as far as we can tell, there is not much more than one can say beyond the obvious decomposition (which follows directly from the definition)
>
> $(X^TX+l*I)^{-1}(X^TX)*\beta+(X^TX+l*I)^{-1}X^T\epsilon$
>
> - We have added a comment about this before the theorem. We also mentioned that 'However, for a random design, we can find a representation that depends on the true covariance $\Sigma$, which may be simpler when $\Sigma$ is simple, e.g., when $\Sigma=I_p$ is isotropic.'
>
> "I think the paper definitely has some merit but the presentation makes it hard to assess it."
>
> We hope that our edits improved the presentation, and that this will make the merits of our paper easier to asses. Please let us know if you have any other feedback and we will do our best to address those issues.

---

### Decision · Program_Chairs · 2019-12-19

**Decision:**

Accept (Spotlight)

**Comment:**

The paper studies theoretical properties of ridge regression, and in particular how to correct for the bias of the estimator.

The reviewers appreciated the contribution and the fact that you updated the manuscript to make it clearer.

I however advise the authors to think about the best way to maximize impact for the ICLR audience, perhaps by providing relevant examples from the ML literature.